# Flow Q-Learning

**Seohong Park** [1]   **Qiyang Li** [1]   **Sergey Levine** [1]

## Abstract

We present flow Q-learning (FQL), a simple and performant offline reinforcement learning (RL) method that leverages an expressive *flow-matching* policy to model arbitrarily complex action distributions in data. Training a flow policy with RL is a tricky problem, due to the iterative nature of the action generation process. We address this challenge by training an expressive *one-step* policy with RL, rather than directly guiding an iterative flow policy to maximize values. This way, we can completely avoid unstable recursive backpropagation, eliminate costly iterative action generation at test time, yet still mostly maintain expressivity. We experimentally show that FQL leads to strong performance across 73 challenging state- and pixel-based OGBench and D4RL tasks in offline RL and offline-to-online RL.

https://seohong.me/projects/fql/

## 1. Introduction

Offline reinforcement learning (RL) enables training an effective decision-making policy from a previously collected dataset without costly environment interactions (Lange et al., 2012; Levine et al., 2020). The essence of offline RL is constrained optimization: the agent must maximize returns while staying within the dataset's state-action distribution (Levine et al., 2020). As datasets have grown larger and more diverse (Collaboration et al., 2024), their behavioral distributions have become more complex and multimodal, and this often necessitates an expressive policy class (Mandlekar et al., 2021) capable of capturing these complex distributions and implementing a more precise behavioral constraint. In this work, we aim to develop a scalable offline RL method by leveraging *flow matching* (Lipman et al., 2023; Liu et al., 2023; Albergo & Vanden-Eijnden, 2023), a simple yet powerful generative modeling technique alternative

[1]University of California, Berkeley. Correspondence to: Seohong Park <seohong@berkeley.edu>.

*Proceedings of the 42$^{nd}$ International Conference on Machine Learning*, Vancouver, Canada. PMLR 267, 2025. Copyright 2025 by the author(s).

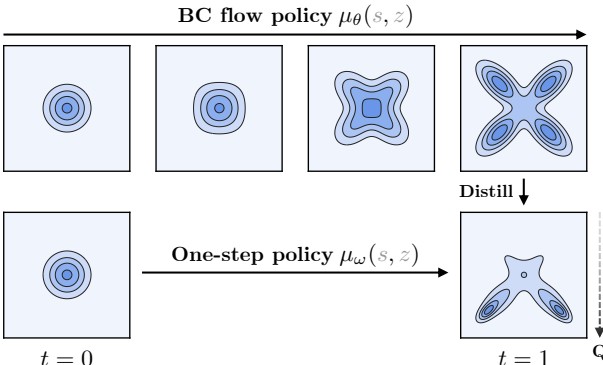

*Figure 1.* **Flow Q-learning.** Flow-matching policies can model complex action distributions, but training an iterative flow policy with RL is challenging. To address this, we train an expressive *one-step* policy : $\mu_\omega(s, z) : \mathcal{S} \times \mathbb{R}^d \to \mathcal{A}$ to maximize Q values, while regularizing it with distillation from a BC flow policy.

to denoising diffusion (Sohl-Dickstein et al., 2015; Ho et al., 2020). By employing an expressive flow policy, we can effectively model the arbitrarily complex action distribution of the dataset and thus enforce an accurate behavioral constraint, which is central to many offline RL algorithms (Nair et al., 2020; Fujimoto & Gu, 2021; Tarasov et al., 2023a).

However, leveraging flow or diffusion models to parameterize policies for offline RL is not a trivial problem. Unlike with simpler policy classes, such as Gaussian policies, there is no straightforward way to train the flow or diffusion policies to maximize a learned value function, due to the iterative nature of these generative models. This is an example of a *policy extraction* problem, which is known to be a key challenge in offline RL in general (Park et al., 2024a). Previous works have devised diverse ways to extract an iterative generative policy from a learned value function, based on weighted regression, reparameterized policy gradient, rejection sampling, and other techniques. While they have shown promising initial results, these extraction schemes are often limited or not necessarily scalable to more complex problems, due to their inherent drawbacks (*e.g.*, unstable backpropagation through time, limited use of samples, and high computational cost; Section 4.1).

In this work, we propose a simple and effective way to leverage an expressive flow policy for offline RL. Our main

idea is to train an iterative flow policy *only* with behavioral cloning (BC). Instead, we train a separate, expressive *one-step* policy that maximizes values while *distilling* from the flow model (Figure 1). By lifting the burden of value maximization from the flow model, we completely avoid the problems associated with steering the iterative process, while fully leveraging the expressivity of the flow model. Moreover, this procedure yields an expressive one-step policy as the output, which eliminates costly iterative flow steps at evaluation time. We call this method **flow Q-learning (FQL)**, which constitutes our main contribution.

FQL is simple: thanks to the simplicity of flow matching (especially compared to denoising diffusion), it can be implemented within a few lines on top of the standard actor-critic framework (Algorithm 1). Yet, FQL is highly effective and efficient. Especially on complex tasks involving highly multimodal action distributions, FQL often leads to significantly better performance than both Gaussian and diffusion policy-based offline RL methods, without requiring iterative flow steps at test time. Moreover, FQL can be directly fine-tuned with online rollouts, often outperforming existing offline-to-online RL methods. We empirically show the effectiveness of FQL on **73** diverse state- and pixel-based tasks across the recently proposed OGBench (Park et al., 2025) and standard D4RL (Fu et al., 2020) benchmarks.

## 2. Preliminaries

**Offline RL.** In this work, we assume a Markov decision process $\mathcal{M}$ (Sutton & Barto, 2005) defined by a tuple $(\mathcal{S}, \mathcal{A}, r, \rho, p)$, where $\mathcal{S}$ is the state space, $\mathcal{A} = \mathbb{R}^d$ is the $d$-dimensional action space, $r(s, a) : \mathcal{S} \times \mathcal{A} \to \mathbb{R}$ is the reward function, $\rho(s) \in \Delta(\mathcal{S})$ is the initial state distribution, and $p(s' \mid s, a) : \mathcal{S} \times \mathcal{A} \to \Delta(\mathcal{S})$ is the transition dynamics distribution, where we denote the set of probability distributions over a space $\mathcal{X}$ as $\Delta(\mathcal{X})$ and use gray to denote placeholder variables. The goal of offline RL is to find the parameter $\theta$ of a policy $\pi_\theta(a \mid s) : \mathcal{S} \to \Delta(\mathcal{A})$ that maximizes the average discounted return $R(\pi_\theta) = \mathbb{E}_{\tau \sim p^{\pi_\theta}(\tau)}[\sum_{h=0}^{H} \gamma^h r(s_h, a_h)]$ from a dataset $\mathcal{D} = \{\tau^{(n)}\}_{n \in \{1,2,\dots,N\}}$ without environment interactions, where $\tau$ denotes a trajectory $(s_0, a_0, \dots, s_H, a_H)$, $\gamma$ denotes a discount factor, and $p^{\pi_\theta}(\tau)$ is defined as $\rho(s_0)\pi_\theta(a_0 \mid s_0)p(s_1 \mid s_0, a_0) \cdots \pi_\theta(a_H \mid s_H)$. In this work, we also consider offline-to-online RL, whose goal is to further fine-tune the offline pre-trained policy with a modest amount of online environment interactions.

**Behavior-regularized actor-critic.**[1] Behavior-regularized actor-critic (Wu et al., 2019; Fujimoto & Gu, 2021; Tarasov

---

[1]Here, we use the term "behavior-regularized actor-critic" to refer to a general framework encompassing a family of approaches, not solely the specific BRAC method (Wu et al., 2019).

et al., 2023a) is one of the simplest (yet effective) offline RL frameworks. In its most basic form, it minimizes the following actor-critic losses:

$$\mathcal{L}_Q(\phi) = \mathbb{E}_{\substack{s,a,r,s' \sim \mathcal{D}, \\ a' \sim \pi_\theta}}[(Q_\phi(s,a) - r - \gamma Q_{\bar{\phi}}(s', a'))^2], \quad (1)$$

$$\mathcal{L}_\pi(\theta) = \mathbb{E}_{s,a \sim \mathcal{D}, a^\pi \sim \pi_\theta}[\underbrace{-Q_\phi(s, a^\pi)}_{\texttt{Q loss}} - \underbrace{\alpha \log \pi(a \mid s)}_{\texttt{BC loss}}], \quad (2)$$

where $Q_\phi(s, a) : \mathcal{S} \times \mathcal{A} \to \mathbb{R}$ is a state-action value function with parameter $\phi$, $Q_{\bar{\phi}}(s, a)$ is a target network (Mnih et al., 2013), $\alpha$ is a hyperparameter that controls the strength of the behavioral cloning (BC) regularizer, and $s, a, r, s' \sim \mathcal{D}$ denotes uniform sampling over the dataset's transition tuples. Intuitively, the critic loss $\mathcal{L}_Q(\phi)$ minimizes the standard Bellman error, while the actor loss $\mathcal{L}_\pi(\theta)$ maximizes values with reparameterized gradients through $a^\pi$, For the actor, the BC loss is additionally applied to prevent the policy from deviating too much from the behavioral policy's distribution. The policy is typically modeled by a Gaussian distribution to enable effective reparameterization. Perhaps surprisingly, despite its simplicity, behavior-regularized actor-critic is one of the most performant frameworks on standard D4RL tasks (Tarasov et al., 2023a). In this work, we build our flow-based offline RL method on a variant of the behavior-regularized actor-critic framework.

**Flow matching.** Flow matching (Lipman et al., 2023; Liu et al., 2023; Albergo & Vanden-Eijnden, 2023) is a simpler alternative to denoising diffusion (Sohl-Dickstein et al., 2015; Ho et al., 2020; Song et al., 2021) for training iterative generative models. Unlike denoising diffusion models, which are based on stochastic differential equations (SDEs), flow models are rooted in deterministic ordinary differential equations (ODEs), which enable significantly simpler training and faster inference, while often achieving better quality (Esser et al., 2024; Lipman et al., 2024).

Given a data distribution $p(x) \in \Delta(\mathbb{R}^d)$ on a $d$-dimensional Euclidean space, flow matching aims to fit the parameter $\theta$ of a time-dependent velocity field $v_\theta(t, x) : [0, 1] \times \mathbb{R}^d \to \mathbb{R}^d$ such that its corresponding *flow* (Lee, 2012) $\psi_\theta(t, x) : [0, 1] \times \mathbb{R}^d \to \mathbb{R}^d$, defined by the unique solution to the ODE

$$\frac{\mathrm{d}}{\mathrm{d}t}\psi_\theta(t, x) = v_\theta(\psi_\theta(t, x)), \quad (3)$$

transforms a simple distribution (*e.g.*, unit Gaussian) at $t = 0$ into the target data distribution $p(x)$ at $t = 1$.

In this work, we consider the simplest variant of flow matching based on linear paths and uniform time sampling (Lipman et al., 2024). The objective is as follows:

$$\min_\theta \ \mathbb{E}_{\substack{x^0 \sim \mathcal{N}(0, I_d), \\ x^1 \sim p(x), \\ t \sim \mathrm{Unif}([0,1])}} \left[ \|v_\theta(t, x^t) - (x^1 - x^0)\|_2^2 \right], \quad (4)$$

where $\mathcal{N}(0, I_d)$ is the $d$-dimensional standard normal distribution, $\text{Unif}([0, 1])$ denotes the uniform distribution over the unit interval, and $x^t = (1 - t)x^0 + tx^1$ is the linear interpolation between $x^0$ and $x^1$. Intuitively, the velocity field $v_\theta$ is trained to match the average direction from randomly sampled $x^0$ and $x^1$. At optimum, this objective produces a vector field that generates the data distribution $p(x)$. At inference time, we generate samples by numerically solving the ODE defined by $v_\theta$. In this work, we use the simplest Euler method, which we find to be sufficient. See Lipman et al. (2024) for further details about flow matching.

**Flow policies.** In this work, we use flow matching to train policies. The most basic flow-matching objective for behavioral cloning is as follows:

$$\mathcal{L}_{\text{Flow}}(\theta) = \mathbb{E}_{\substack{s, a=x^1 \sim \mathcal{D}, \\ x^0 \sim \mathcal{N}(0, I_d), \\ t \sim \text{Unif}([0,1])}} \left[ \|v_\theta(t, s, x^t) - (x^1 - x^0)\|_2^2 \right], \tag{5}$$

where $v_\theta(t, s, x) : [0, 1] \times \mathcal{S} \times \mathbb{R}^d \to \mathbb{R}^d$ is a state- and time-dependent vector field with parameter $\theta$. Recall that $\mathcal{A}$ is defined as $\mathbb{R}^d$, and flow matching happens in the action space. The state-dependent vector field generates a state-dependent flow $\psi_\theta(t, s, x) : [0, 1] \times \mathcal{S} \times \mathbb{R}^d \to \mathbb{R}^d$, which serves as a policy. For $s \in \mathcal{S}$ and $z \in \mathbb{R}^d$, we simply denote the ODE's output $\psi_\theta(1, s, z)$ by $\mu_\theta(s, z)$. Intuitively, $\mu_\theta$ maps the noise $z = x^0$ (sampled from the standard normal distribution) to the action $a = \mu_\theta(s, z)$ by the ODE.

**Notational warning:** Note that $\mu_\theta(s, z)$ is a *deterministic function* from $\mathcal{S} \times \mathbb{R}^d$ to $\mathcal{A}$, but serves as a *stochastic policy* from $\mathcal{S}$ to $\mathcal{A}$ due to the stochasticity of $z \sim \mathcal{N}(0, I_d)$. We denote the corresponding induced stochastic policy as $\pi_\theta(a \mid s)$, and loosely refer to both $\mu_\theta$ and $\pi_\theta$ as "policies."

## 3. Flow Q-Learning

We now introduce our method for effective data-driven decision-making, **flow Q-learning (FQL)**. Our desiderata are twofold: we want to leverage an expressive flow-matching policy to deal with complex behavioral action distributions; we also want to keep the method as simple as possible so that practitioners can easily implement and use it.

**Naïve approach.** Perhaps the simplest way to train a flow policy for offline RL is to replace the BC loss with a flow-matching loss (Equation (5)) in the behavior-regularized actor-critic framework (Equation (2)). Formally, this naïve approach minimizes the actor loss $\mathcal{L}_\pi(\theta)$ defined by

$$\mathcal{L}_\pi(\theta) = \underbrace{\mathbb{E}_{s \sim \mathcal{D}, a^\pi \sim \pi_\theta}[-Q_\phi(s, a^\pi)]}_{\text{Q loss}} + \underbrace{\alpha \mathcal{L}_{\text{Flow}}(\theta)}_{\text{BC loss}}. \tag{6}$$

Intuitively, the corresponding flow policy $\pi_\theta$ is "steered" to maximize the value function while minimizing the BC loss. This is analogous to Diffusion-QL (Wang et al., 2023)

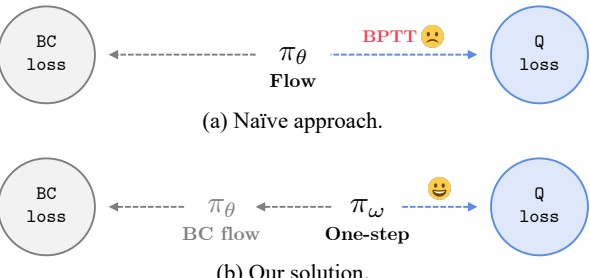

(a) Naïve approach.

(b) Our solution.

*Figure 2.* **The idea.** Offline RL is essentially a tug-of-war between behavioral regularization and value maximization. *(a)* Naïvely doing this with a flow policy involves costly and unstable backpropagation through time (BPTT). *(b)* We resolve this by training a separate *one-step* policy, which maximizes values without BPTT while being regularized by a distillation loss from a BC flow policy.

for diffusion policies. However, unlike the Gaussian case, the flow or diffusion objective requires *backpropagation through time* in the Q loss (Equation (6)) due to the recursion in numerical ODE solvers (*e.g.*, the Euler method) (Figure 2a). Unfortunately, this is often unstable and costly in practice, potentially leading to suboptimal performance, as we will show in our experiments.

**Solution.** Our main idea is to **not** steer the original flow policy at all. Instead, we will train the flow policy only with the BC loss, and train a separate expressive *one-step* policy to maximize the value function while regularizing it by a *distillation* loss from the full BC flow policy. Since the one-step policy does not involve any iterative procedures, we can completely avoid backpropagation through time in the Q loss (Equation (6)). We call this idea **one-step guidance**.

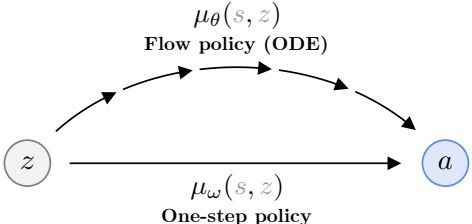

*Figure 3.* **One-step policy.** The one-step policy $\mu_\omega$ learns the *direct* mapping from $z$ to $a$ of the flow policy $\mu_\theta$, while simultaneously maximizing values (this part is omitted in the figure).

More formally, we train a flow policy $\mu_\theta(s, z)$ only with the BC flow-matching loss (Equation (5)). Alongside, we train a one-step prediction model $\mu_\omega(s, z) : \mathcal{S} \times \mathbb{R}^d \to \mathcal{A}$ with parameter $\omega$, whose main role is to learn the *direct* mapping from noise $z$ to the output action of the full ODE flow policy $a = \mu_\theta(s, z)$, while simultaneously maximizing

**Algorithm 1** Flow Q-Learning (FQL)

**function** $\mu_\theta(s, z)$        ▷ BC flow policy
    **for** $t = 0, 1, \ldots, M - 1$ **do**
         $z \leftarrow z + v_\theta(t/M, s, z)/M$    ▷ Euler method
    **return** $z$

**while** not converged **do**
    Sample batch $\{(s, a, r, s')\} \sim \mathcal{D}$
    ▷ Train critic $Q_\phi$
    $z \sim \mathcal{N}(0, I_d)$
    $a' \leftarrow \mu_\omega(s', z)$
    Update $\phi$ to minimize $\mathbb{E}[(Q_\phi(s, a) - r - \gamma Q_{\bar{\phi}}(s', a'))^2]$

    ▷ Train vector field $v_\theta$ in BC flow policy $\pi_\theta$
    $x^0 \sim \mathcal{N}(0, I_d)$
    $x^1 \leftarrow a$
    $t \sim \mathrm{Unif}([0, 1])$
    $x^t \leftarrow (1 - t)x^0 + tx^1$
    Update $\theta$ to minimize $\mathbb{E}[\|v_\theta(t, s, x^t) - (x^1 - x^0)\|_2^2]$

    ▷ Train one-step policy $\pi_\omega$
    $z \sim \mathcal{N}(0, I_d)$
    $a^\pi \leftarrow \mu_\omega(s, z)$
    Update $\omega$ to minimize $\mathbb{E}[-Q_\phi(s, a^\pi) + \alpha\|a^\pi - \mu_\theta(s, z)\|_2^2]$
**return** One-step policy $\pi_\omega$

values (Figure 3). The distillation loss is defined as follows:

$$\mathcal{L}_{\mathrm{Distill}}(\omega) = \mathbb{E}_{\substack{s \sim \mathcal{D}, \\ z \sim \mathcal{N}(0, I_d)}} \left[\|\mu_\omega(s, z) - \mu_\theta(s, z)\|_2^2\right]. \quad (7)$$

Recall that $\mu_\theta(s, z)$ denotes the output of the ODE defined by the vector field $v_\theta$ (Section 2). Importantly, we note that it is possible to train an *expressive* one-step model that generates high-quality samples with distillation losses (Liu et al., 2023; 2024; Li et al., 2024a; Ding et al., 2024b; Frans et al., 2025).

We are now ready to describe the complete objective of our method, **flow Q-learning (FQL)**. FQL has three components: critic $Q_\phi(s, a)$, BC flow policy $\mu_\theta(s, z)$, and one-step policy $\mu_\omega(s, z)$. First, as discussed above, the BC flow policy is trained *only* with the BC flow-matching loss (Equation (5)). The critic is trained with the original critic loss of behavior-regularized actor-critic (Equation (1)), except that we use the one-step policy $\pi_\omega$ in place of $\pi_\theta$. Finally, the one-step policy is trained with the following actor loss:

$$\mathcal{L}_\pi(\omega) = \underbrace{\mathbb{E}_{s \sim \mathcal{D}, a^\pi \sim \pi_\omega}[-Q_\phi(s, a^\pi)]}_{\texttt{Q loss}} + \underbrace{\alpha \mathcal{L}_{\mathrm{Distill}}(\omega)}_{\texttt{"BC" loss}}. \quad (9)$$

Similar to the naïve flow actor loss above (Equation (6)), this objective maximizes both the Q and BC losses with a hyperparameter $\alpha$. However, it does not involve backpropagation over time as $\pi_\omega$ is a one-step policy. Note also that the distillation loss now serves as a behavioral regularizer based on the BC flow policy (Figure 2b). The output of this algorithm is the one-step policy $\pi_\omega$, which is what is deployed at test time. We provide a pseudocode for FQL in Algorithm 1, in

**Remark:** Connection to Wasserstein Regularization

Our distillation loss in Equation (7) has an intriguing connection to *Wasserstein* behavioral regularization. Let $\xi$ be a random variable following the $d$-dimensional standard normal distribution, $\mathcal{N}(0, I_d)$. For $s \in \mathcal{S}$, let $\pi_\theta(s), \pi_\omega(s) \in \Delta(\mathcal{A})$ be the push-forward distributions of $\xi$ by $\mu_\theta(s, \cdot)$ and $\mu_\omega(s, \cdot)$, respectively. Then, the distillation loss in Equation (7) is an upper bound on the squared 2-Wasserstein distance between $\pi_\omega(s)$ and $\pi_\theta(s)$:

$$\mathcal{L}_{\mathrm{Distill}}(\omega) = \mathbb{E}_{\substack{s \sim \mathcal{D}, \\ z \sim \mathcal{N}(0, I_d)}} \left[\|\mu_\omega(s, z) - \mu_\theta(s, z)\|_2^2\right]$$

$$\geq \mathbb{E}_{s \sim \mathcal{D}} \left[\inf_{\lambda \in \Lambda(\pi_\omega, \pi_\theta)} \mathbb{E}_{x, y \sim \lambda}[\|x - y\|_2^2]\right]$$

$$= \mathbb{E}_{s \sim \mathcal{D}} \left[W_2(\pi_\omega, \pi_\theta)^2\right], \quad (8)$$

where $\Lambda(\pi_\omega, \pi_\theta)$ denotes the set of coupling distributions of $\pi_\omega$ and $\pi_\theta$, and $W_2$ denotes the 2-Wasserstein distance with the Euclidean metric in the action space.

*Table 1.* **Behavioral regularizers in offline RL.**

| Offline RL Method | Behavioral Regularizer | Metric-Aware? |
|---|---|---|
| TD3+BC | $D_{\mathrm{KL}}$ | ✗ |
| AWAC | $D_{\mathrm{KL}}$ | ✗ |
| CQL | $\chi^2$ | ✗ |
| FQL (ours) | $W_2^2$ | ○ |

Hence, the BC term in the FQL actor loss (Equation (9)) can be interpreted as an upper bound on the squared 2-Wasserstein distance between the current policy $\pi_\omega$ and the data-collecting policy approximated by $\pi_\theta$. This Wasserstein regularizer is analogous to the KL behavioral regularizer in TD3+BC (Fujimoto & Gu, 2021) and AWAC (Nair et al., 2020), and the $\chi^2$ behavioral regularizer in CQL (Kumar et al., 2020; Garg et al., 2023). However, unlike the KL and $\chi^2$ divergences, which are (in principle) invariant and agnostic to any metric structures,[a] our 2-Wasserstein distance is *aware* of the metric structure over actions (which we impose as the Euclidean distance) (Table 1). This metric-aware property potentially incorporates a better inductive bias about the similarity between actions, akin to how Wasserstein distances improve upon metric-agnostic divergences in other contexts in machine learning (Arjovsky et al., 2017; Park et al., 2024b).

    [a]While the original KL and $\chi^2$ divergences are entirely metric-agnostic, this property may be lost in practice with variational approximation (*e.g.*, with a Gaussian parameterization).

which $M$ denotes the number of steps for the Euler method, and describe the full implementation details in Appendix B.

**Why is it a good idea?** FQL has three benefits. First, it leverages reparameterized policy gradient (*i.e.*, directly maximizing the Q function with gradients through $a^\pi$), which is known to be one of the most effective policy extraction methods (Park et al., 2024a), while entirely avoiding unstable and costly backpropagation through time. We will revisit this point in more detail in Section 4.1, and empirically show its effectiveness through our experiments (Section 5). Second, FQL yields an efficient one-step policy as the output, which eliminates iterative flow generation processes at inference time, while maintaining most of the expressivity of the full flow model (Liu et al., 2023; Frans et al., 2025). Third, FQL is easy-to-implement and easy-to-tune: thanks to the simplicity of flow-matching, it can be implemented in a few lines on top of the standard behavior-regularized actor-critic framework, and has only one major hyperparameter $\alpha$, without requiring tuning a noise schedule.

# 4. Prior Work

**Offline RL and offline-to-online RL.** The goal of offline RL is to train a policy using only previously collected data. Hundreds of offline RL methods and techniques have been proposed so far, and many of them are based on a single central idea: maximizing the return while minimizing a discrepancy measure between the state-action distribution of the dataset and that of the learned policy (Levine et al., 2020; Sikchi et al., 2024). Previous works have implemented this high-level objective in diverse ways through behavioral regularization (Nair et al., 2020; Fujimoto & Gu, 2021; Tarasov et al., 2023a), conservatism (Kumar et al., 2020), in-sample maximization (Kostrikov et al., 2022; Xu et al., 2023; Garg et al., 2023), out-of-distribution detection (Yu et al., 2020; Kidambi et al., 2020; An et al., 2021; Nikulin et al., 2023), dual RL (Lee et al., 2021a; Sikchi et al., 2024), and generative modeling (Chen et al., 2021; Janner et al., 2021; 2022). After finishing offline RL training, we can further fine-tune the policy with additional online rollouts. This setting is often referred to as offline-to-online RL, for which several techniques have been proposed (Lee et al., 2021b; Song et al., 2023; Nakamoto et al., 2023; Ball et al., 2023; Yu & Zhang, 2023). Our method, FQL, is mainly designed for offline RL, but we show that it can also be directly fine-tuned with online rollouts without any algorithmic changes.

**RL with diffusion and flow models.** Motivated by the recent successes of iterative generative modeling techniques, such as denoising diffusion (Sohl-Dickstein et al., 2015; Ho et al., 2020; Dhariwal & Nichol, 2021) and flow matching (Lipman et al., 2023; Esser et al., 2024), researchers have developed diverse ways to integrate them into RL. Previous works have applied iterative generative models to

planning and hierarchical learning (Janner et al., 2022; Ajay et al., 2023; Zheng et al., 2023; Liang et al., 2023; Li et al., 2023; Suh et al., 2023; Venkatraman et al., 2024; Chen et al., 2024a), world modeling and data augmentation (Lu et al., 2023a; Ding et al., 2024c; Jackson et al., 2024; Alonso et al., 2024), exploration (Mazoure et al., 2019; Ren et al., 2025), and policy modeling (Section 4.1). Our method belongs to the third category, where we model a policy with an expressive flow network to capture the arbitrarily complex distribution of the behavioral policy.

## 4.1. How Have Previous Works Trained Diffusion and Flow Policies with RL?

Various approaches have been proposed for training diffusion or flow policies with RL. In this section, we provide an in-depth review of these methods, discuss their advantages and limitations, and explain how FQL relates to prior work. Prior methods can be categorized into several groups based on their *policy extraction* strategies (Park et al., 2024a).

**(1) Weighted behavioral cloning.** One straightforward approach to modulating a diffusion or flow policy is to assign *weights* to transition samples based on the corresponding learned values. The most basic form uses advantage-weighted regression (AWR) (Peters & Schaal, 2007; Peng et al., 2019; Nair et al., 2020) with the following objective:

$$\max_\theta \quad \mathbb{E}_{s,a\sim\mathcal{D}} \left[ e^{\alpha(Q(s,a)-V(s))} \mathcal{L}_{\text{Flow}}(\theta) \right], \quad (10)$$

where $\alpha$ is an inverse temperature hyperparameter, and $Q(s,a) : \mathcal{S} \times \mathcal{A} \to \mathbb{R}$ and $V(s) : \mathcal{S} \to \mathbb{R}$ are state-action and state value functions, respectively (Sutton & Barto, 2005). For diffusion policies, $\mathcal{L}_{\text{Flow}}(\theta)$ is replaced with a diffusion loss. Intuitively, this objective makes the policy selectively clone transitions with high advantages. Among previous works, QGPO (Lu et al., 2023b), EDP (Kang et al., 2023), QVPO (Ding et al., 2024a), and QIPO (Zhang et al., 2025) are mainly based on weighted behavioral cloning.

Weighted behavioral cloning is simple and easy to implement. However, it is known to be one of the least effective policy extraction methods (Fu et al., 2022; Park et al., 2024a), due to the small number of effective samples and limited expressivity.[2] In our experiments, we empirically show that weighted behavioral cloning generally leads to subpar performance, especially on complex tasks.

**(2) Reparameterized policy gradient.** Another popular approach to guide an iterative generative model is to directly maximize the value function $Q(s,a)$ with reparameterized gradients, while regularizing it with a flow or diffusion loss, as in Equation (6). Among previous approaches, Diffusion-QL (Wang et al., 2023), DiffCPS (He et al., 2023), Consistency-AC (Ding & Jin, 2024), SRDP (Ada et al.,

---

[2]See Park et al. (2024a) for further discussions.

2024), and EQL (Zhang et al., 2024) implement this scheme with backpropagation through time.

Reparameterized policy gradient is known to be one of the most effective policy extraction methods for Gaussian policies (Park et al., 2024a). However, when naïvely applied to iterative generative models, it requires backpropagation through time (Equation (9)), which often incurs stability issues and leads to suboptimal performance (Section 5).

**(3) Rejection sampling.** The third category is rejection sampling. Instead of adjusting the parameter of the generative model, we can sample $N$ actions from a *fixed* BC policy, and select the action that has the highest value. In other words, we treat the following formula as a policy:

$$\underset{a_1,\ldots,a_N \,:\, a_i \sim \pi^\beta}{\arg\max} \quad Q(s, a_i), \qquad (11)$$

where $\pi^\beta$ is a BC policy trained by a flow or diffusion objective. Among previous works, SfBC (Chen et al., 2023), IDQL (Hansen-Estruch et al., 2023), and AlignIQL (He et al., 2024) are based on (variants of) rejection sampling.

Rejection sampling is simple and stable. However, it requires querying the policy and value function $N$ times at *every* environment step during inference (and possibly during training as well, depending on the method). This can be prohibitive with larger models or a larger number of samples.

**(4) Others.** Besides these three major categories, other techniques have also been proposed to guide a diffusion policy to maximize the learned value function, based on some combination of the above strategies (Mao et al., 2024), action gradients (Yang et al., 2023; Psenka et al., 2024; Li et al., 2024b; Mark et al., 2024; Fang et al., 2025), bi-level MDPs (Ren et al., 2025), value alignment (Chen et al., 2024c), and implicit Q-learning (Chen et al., 2024b;d).

**Contextualizing FQL in prior work.** Our approach, FQL, falls into the second category, reparameterized policy gradient, which is known to be one of the most effective policy extraction schemes (Park et al., 2024a). However, unlike the previous methods discussed above in the same category, which use backpropagation through time, we entirely bypass recursive backpropagation by only steering the one-step policy to maximize values (Equation (9)), while training the flow policy solely with the BC loss. Among previous works, Consistency-AC (Ding & Jin, 2024), SRPO (Chen et al., 2024b), and DTQL (Chen et al., 2024d) also employ distillation, and in particular, Consistency-AC (Ding & Jin, 2024) shares a conceptually similar high-level objective to our method (but with consistency models instead of direct one-step distillation). However, they either still use backpropagation through time (Ding & Jin, 2024) or are based on implicit Q-learning (Kostrikov et al., 2022), which is known to be less effective than actor-critic learning (Tarasov et al., 2023a). In contrast, we train a *one-step* policy within a more

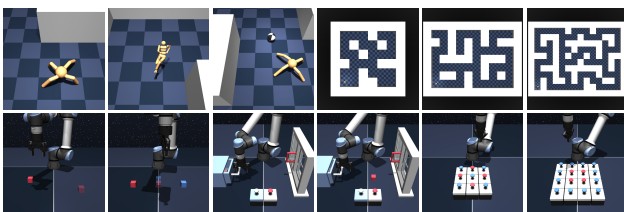

*Figure 4.* **OGBench tasks.**

effective actor-critic framework, with no backpropagation through time. In our experiments, we empirically show that our approach leads to significantly better performance than previous distillation-based methods (Consistency-AC and SRPO) as well as other policy extraction schemes.

## 5. Experiments

In this section, we empirically evaluate the performance of FQL, comparing it to previous offline RL and offline-to-online RL approaches on a variety of challenging tasks. We also provide extensive analyses and ablations on policy extraction strategies and FQL's design choices.

### 5.1. Experimental Setup

**Benchmarks.** We use the recently proposed **OGBench** task suite (Park et al., 2025) as the main benchmark (Figure 4). OGBench provides a number of diverse, challenging tasks across robotic locomotion and manipulation, with both state and pixel observations, where these tasks are generally more challenging than standard D4RL tasks (Fu et al., 2020), which have been saturated as of 2025 (Tarasov et al., 2023a; Rafailov et al., 2024; Park et al., 2024a). While OGBench was originally designed for benchmarking offline goal-conditioned RL, we use its reward-based single-task variants ("-singletask") to make it compatible with standard reward-maximizing offline RL algorithms. We employ 5 locomotion and 5 manipulation environments where each environment provides 5 separate tasks, bringing the total to **50** state-based OGBench tasks. In addition, we consider 5 diverse OGBench **visual** manipulation tasks to challenge the agent's ability to handle $64 \times 64 \times 3$-sized image observations. Finally, we also employ relatively challenging 6 antmaze and 12 adroit tasks from the **D4RL** benchmark.

**Methods.** For our offline RL experiments, we use the following 9 recent methods as representative examples of a variety of algorithm types and policy extraction strategies.

**(1) Gaussian policies.** For standard offline RL methods that use Gaussian policies, we consider BC, IQL (Kostrikov et al., 2022), and ReBRAC (Tarasov et al., 2023a). In particular, ReBRAC is known to achieve state-of-the-art performance on many D4RL tasks (Tarasov et al., 2023b), and is

*Table 2.* **Offline RL results.** FQL achieves the best or near-best performance on most of the **73** diverse, challenging benchmark tasks. The performances are averaged over **8** seeds (**4** seeds for pixel-based tasks), but the cells without the "$\pm$" sign indicate that the numbers are taken from prior works (Tarasov et al., 2023b; Hansen-Estruch et al., 2023; Chen et al., 2024b). See Table 3 for the full results.

| | Gaussian Policies | | | Diffusion Policies | | | Flow Policies | | | |
|---|---|---|---|---|---|---|---|---|---|---|
| Task Category | BC | IQL | ReBRAC | IDQL | SRPO | CAC | FAWAC | FBRAC | IFQL | FQL |
| OGBench antmaze-large-singletask (**5** tasks) | $11_{\pm1}$ | $53_{\pm3}$ | $\mathbf{81}_{\pm5}$ | $21_{\pm5}$ | $11_{\pm4}$ | $33_{\pm4}$ | $6_{\pm1}$ | $60_{\pm6}$ | $28_{\pm5}$ | $79_{\pm3}$ |
| OGBench antmaze-giant-singletask (**5** tasks) | $0_{\pm0}$ | $4_{\pm1}$ | $\mathbf{26}_{\pm8}$ | $0_{\pm0}$ | $0_{\pm0}$ | $0_{\pm0}$ | $0_{\pm0}$ | $4_{\pm4}$ | $3_{\pm2}$ | $9_{\pm6}$ |
| OGBench humanoidmaze-medium-singletask (**5** tasks) | $2_{\pm1}$ | $33_{\pm2}$ | $22_{\pm8}$ | $1_{\pm0}$ | $1_{\pm1}$ | $53_{\pm8}$ | $19_{\pm1}$ | $38_{\pm5}$ | $60_{\pm14}$ | $58_{\pm5}$ |
| OGBench humanoidmaze-large-singletask (**5** tasks) | $1_{\pm0}$ | $2_{\pm1}$ | $2_{\pm1}$ | $1_{\pm0}$ | $0_{\pm0}$ | $0_{\pm0}$ | $0_{\pm0}$ | $2_{\pm0}$ | $\mathbf{11}_{\pm2}$ | $4_{\pm2}$ |
| OGBench antsoccer-arena-singletask (**5** tasks) | $1_{\pm0}$ | $8_{\pm2}$ | $0_{\pm0}$ | $12_{\pm4}$ | $1_{\pm0}$ | $2_{\pm4}$ | $12_{\pm0}$ | $16_{\pm1}$ | $33_{\pm6}$ | $60_{\pm2}$ |
| OGBench cube-single-singletask (**5** tasks) | $5_{\pm1}$ | $83_{\pm3}$ | $91_{\pm2}$ | $\mathbf{95}_{\pm2}$ | $80_{\pm5}$ | $85_{\pm9}$ | $81_{\pm4}$ | $79_{\pm7}$ | $79_{\pm2}$ | $\mathbf{96}_{\pm1}$ |
| OGBench cube-double-singletask (**5** tasks) | $2_{\pm1}$ | $7_{\pm1}$ | $12_{\pm1}$ | $15_{\pm6}$ | $2_{\pm1}$ | $6_{\pm2}$ | $5_{\pm2}$ | $15_{\pm3}$ | $14_{\pm3}$ | $\mathbf{29}_{\pm2}$ |
| OGBench scene-singletask (**5** tasks) | $5_{\pm1}$ | $28_{\pm1}$ | $41_{\pm3}$ | $46_{\pm3}$ | $20_{\pm1}$ | $40_{\pm7}$ | $30_{\pm3}$ | $45_{\pm5}$ | $30_{\pm3}$ | $56_{\pm2}$ |
| OGBench puzzle-3x3-singletask (**5** tasks) | $2_{\pm0}$ | $9_{\pm1}$ | $21_{\pm1}$ | $10_{\pm2}$ | $18_{\pm1}$ | $19_{\pm0}$ | $6_{\pm2}$ | $14_{\pm4}$ | $19_{\pm1}$ | $\mathbf{30}_{\pm1}$ |
| OGBench puzzle-4x4-singletask (**5** tasks) | $0_{\pm0}$ | $7_{\pm1}$ | $14_{\pm1}$ | $\mathbf{29}_{\pm3}$ | $10_{\pm3}$ | $15_{\pm3}$ | $1_{\pm0}$ | $13_{\pm1}$ | $25_{\pm5}$ | $17_{\pm2}$ |
| D4RL antmaze (**6** tasks) | $17$ | $57$ | $78$ | $79$ | $74$ | $30_{\pm3}$ | $44_{\pm3}$ | $64_{\pm7}$ | $65_{\pm7}$ | $\mathbf{84}_{\pm3}$ |
| D4RL adroit (**12** tasks) | $48$ | $53$ | $\mathbf{59}$ | $52_{\pm1}$ | $51_{\pm1}$ | $43_{\pm2}$ | $48_{\pm1}$ | $50_{\pm2}$ | $52_{\pm1}$ | $52_{\pm1}$ |
| Visual manipulation (**5** tasks) | $-$ | $42_{\pm4}$ | $60_{\pm2}$ | $-$ | $-$ | $-$ | $-$ | $22_{\pm2}$ | $50_{\pm5}$ | $\mathbf{65}_{\pm2}$ |

[1] Due to the high computational cost of pixel-based tasks, we selectively benchmark 5 methods that achieve strong performance on state-based OGBench tasks.

the closest Gaussian baseline to FQL in that both are based on behavior-regularized actor-critic (Section 2).

**(2) Diffusion policies.** For diffusion policy-based offline RL methods, we consider IDQL (Hansen-Estruch et al., 2023), SRPO (Chen et al., 2024b), and Consistency-AC (CAC) (Ding & Jin, 2024). IDQL is based on rejection sampling, and SRPO and CAC are based on policy distillation, as in FQL. In particular, CAC is the closest diffusion baseline to FQL, in that they both train distillation policies within the behavior-regularized actor-critic framework, although CAC still employs backpropagation through time (but with fewer steps) and is based on consistency models rather than direct one-step distillation.

**(3) Flow policies.** Since there are currently only a few prior methods that explicitly employ flow policies (Zhang et al., 2025), we consider flow variants of previous methods to cover the three main policy extraction schemes discussed in Section 4.1. Flow advantage-weighted actor-critic (FAWAC) is a flow variant of AWAC (Nair et al., 2020), which uses AWR (Equation (10)) as the policy learning objective, conceptually similar to QIPO (Zhang et al., 2025). Flow behavior-regularized actor-critic (FBRAC) is the flow counterpart of Diffusion-QL (DQL) (Wang et al., 2023) based on the naïve Q loss with backpropagation through time (Equation (6)). Implicit flow Q-learning (IFQL) is the flow counterpart of IDQL based on rejection sampling (Equation (11)). Notably, FAWAC and FBRAC are different from our method (FQL) *only* by their policy extraction strategies while sharing the exact same architectures and implementations, and thus can provide controlled ablation results on our distillation-based policy extraction scheme.

For offline-to-online RL experiments, we consider three prior offline RL methods (IQL, ReBRAC, and IFQL) that support fine-tuning and achieve strong performance. Addi-

tionally, we consider two performant methods specifically designed for data-driven online RL, Cal-QL (Nakamoto et al., 2023) and RLPD (Ball et al., 2023).

**Evaluation.** For offline RL, we evaluate the performance of methods after a fixed number of gradient steps; in particular, we do *not* report the best performance across different evaluation epochs as it may bias results (Tarasov et al., 2023b). To ensure fair comparisons, we *individually* tune hyperparameters of the baselines with similar amounts of training budget (Appendix E.2), and use the same network size and discount factor, unless otherwise stated. We use **8** seeds for state-based tasks and **4** seeds for pixel-based tasks, and present standard deviations after "$\pm$" in tables and $95\%$ bootstrap confidence intervals as shaded areas in plots, unless otherwise mentioned. In tables, we denote values at or above $95\%$ of the best performance in bold, following OG-Bench (Park et al., 2025). We refer to Appendix E for the full training and evaluation details.

### 5.2. Results and Q&As

We present our results via the following Q&As.

**Q: How good is FQL for offline RL?**

**A:** FQL achieves the best or near-best performance on most tasks, especially in complex manipulation environments.

Table 2 summarizes the aggregated benchmarking result on a total of 73 state- or pixel-based offline RL tasks across robotic locomotion and manipulation. We find that FQL generally achieves better performance than previous methods, including ones based on Gaussian and diffusion policies. In particular, FQL leads to consistently better performance than its closest diffusion baseline (CAC), and often significantly outperforms its closest Gaussian baseline (ReBRAC) especially on manipulation tasks, which feature highly mul-

timodal distributions. We also highlight that FQL achieves the best performance of **84%** on one of the hardest tasks in the D4RL benchmark, `antmaze-large-play` (Table 3).

**Q: Can't I just use existing policy extraction schemes?**

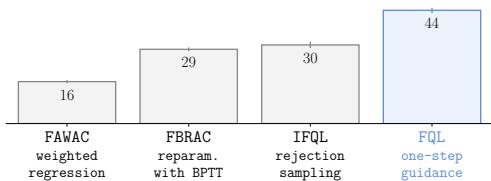

*Figure 5.* **Policy extraction is important.** The bars above compare the performances of different policy extraction methods averaged over the 50 state-based OGBench tasks in Table 2.

**A:** You can, but previous policy extraction schemes generally lead to (often *much*) worse performance.

This can be seen by comparing the performances of FQL and {FAWAC, FBRAC, IFQL}, which are the closest flow-based baselines to FQL, but with different policy extraction mechanisms. In particular, FBRAC is **exactly** the same as FQL except that it uses backpropagation through time. We emphasize again that these baselines are implemented on the same codebase, use the same architecture, and are individually tuned for each environment (Table 6). Figure 5 compares their offline RL performances aggregated over the 50 state-based OGBench tasks in Table 2. The results show that policy extraction alone can significantly affect performance, consistent with findings in Gaussian policies (Park et al., 2024a). The results also indicate that our one-step guidance is the most effective, significantly outperforming the other previous extraction strategies (Section 4.1).

**Q: Can FQL be fine-tuned with online rollouts?**

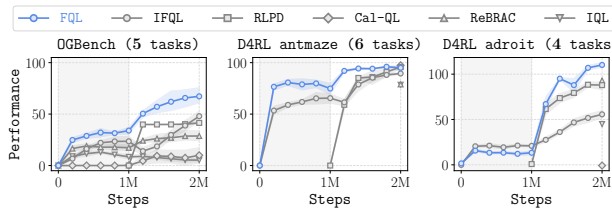

*Figure 6.* **Offline-to-online RL results (8 seeds).** Fine-tuning starts at 1M. The D4RL results of Cal-QL, ReBRAC, and IQL are taken from Tarasov et al. (2023b). See Figure 12 for the full plots.

**A:** Yes, FQL can be directly fine-tuned without any modifications, and often significantly outperforms previous methods.

Specifically, we can fine-tune FQL simply by adding new online transitions to the dataset $\mathcal{D}$, while continuing to train all networks using the same objective as in offline training. To show how effective FQL is for fine-tuning, we evaluate it on 5 representative OGBench tasks across different categories

(Table 4) as well as the 10 D4RL `antmaze` and `adroit` tasks used by Tarasov et al. (2023b). Figure 6 shows the training curves of FQL and previous approaches on these 15 tasks, where online fine-tuning starts at 1M gradient steps (see Figure 12 and Table 4 for the full results). The results show that FQL achieves the best fine-tuning performance compared to both previous offline RL approaches (including IFQL, the strongest flow-based baseline) and methods specifically designed for online fine-tuning (Cal-QL and RLPD).

**Q: What are the important hyperparameters of FQL?**

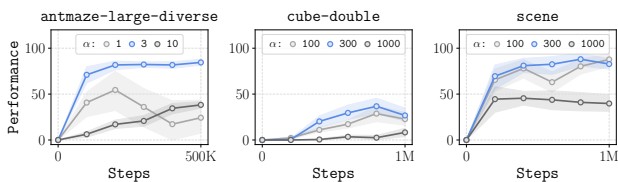

*Figure 7.* **The BC coefficient $\alpha$ needs to be tuned.** The plots show how different values of $\alpha$ affect offline RL performance.

**A:** The most important hyperparameter is the BC coefficient.

Figure 7 shows the ablation results of the BC coefficient $\alpha$ on three tasks. This hyperparameter needs to be tuned for each environment based on the suboptimality of the dataset, as is typical for most offline RL methods (Tarasov et al., 2023b; Park et al., 2024a). Other than $\alpha$, the default hyperparameters of FQL work well, although tuning some additional hyperparameters (*e.g.*, target value aggregation described in Appendix B) can slightly boost performance on some tasks. We provide an extensive ablation study on a total of 4 factors of FQL in Appendix C.

**Q: Do I need to tune flow-related hyperparameters?**

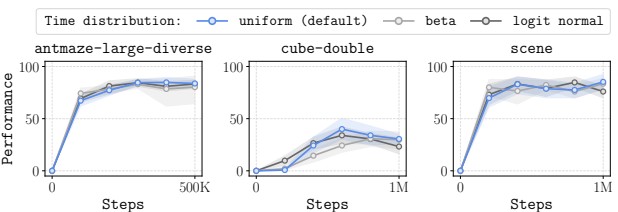

*Figure 8.* **You can just use the uniform time distribution.** FQL's performance is generally robust to flow-related hyperparameters.

**A:** No, in general.

For example, Figure 8 shows how the time sampling distribution for flow matching affects performance, where we consider the uniform distribution, $\mathrm{Unif}([0, 1])$ (default), the beta distribution used by Black et al. (2024), and the logit normal distribution used by Esser et al. (2024). The results suggest that time distributions matter only marginally, and the simplest uniform distribution is often sufficient to

achieve the best performance. Similarly, we find that the performance is generally robust to the number of flow steps (the default is 10), as long as it is not too small (see Appendix C).

### Q: How fast is FQL?

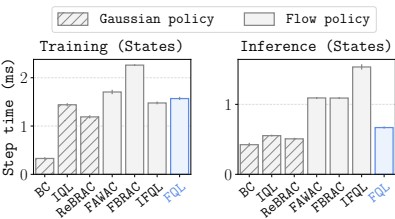

*Figure 9.* **Run time comparison on** `cube-double`.

**A:** FQL is one of the fastest flow-based offline RL methods.

Figure 9 shows that, in terms of both training and inference costs, FQL is only slightly slower than Gaussian policy-based offline RL methods, while being faster than most flow-based baselines. See Figure 11 for the detailed comparison results.

### Q: Are flow policies better than diffusion policies?

**A:** Maybe, but we do **not** make such a claim in this paper.

The main contribution of this paper is our *policy extraction* scheme (one-step guidance), not just the use of flow matching itself. Although we show that one-step guidance combined with flow matching (*i.e.*, FQL) achieves better performance than previous policy extraction schemes for diffusion and flow policies (Table 2), we believe it is possible to apply our one-step guidance to diffusion policies with appropriate modifications to convert SDEs to ODEs (Song et al., 2021) to achieve similar performance, given the equivalence between the two frameworks (Gao et al., 2024). Nevertheless, flow matching has one arguably clear advantage over denoising diffusion: it is *much* simpler to implement!

## 6. Closing Remarks

We presented flow Q-learning (FQL), a simple and performant offline RL method that leverages an expressive flow policy and reparameterized policy gradient, without suffering from backpropagation through time. We showed that FQL generally leads to the best performance on challenging tasks across robotic locomotion and manipulation, offline RL and offline-to-online RL, as well as state- and pixel-based settings. FQL, however, is not perfect; see Appendix A for the limitations of FQL.

As a closing remark, we would like to reiterate one particularly appealing property of FQL — **simplicity**: one small algorithm box (Algorithm 1) essentially captures the entire training objectives of FQL (modulo minor details), *including* all of flow matching, iterative sampling, and value learn-

ing. Given that offline RL is notoriously sensitive to implementation details in general (Tarasov et al., 2023b), we believe proposing a simple yet performant method is a particularly important contribution to the community. We hope that FQL, with our clean, open-source implementation, spurs future research in scalable offline RL algorithms.

## Acknowledgments

We thank Chongyi Zheng for noticing an issue in our initial implementation. This work was partly supported by the Korea Foundation for Advanced Studies (KFAS), AFOSR FA9550-22-1-0273, and ONR N00014-20-1-2383. This research used the Savio computational cluster resource provided by the Berkeley Research Computing program at UC Berkeley. Some figures in this work use Twemoji, an open-source emoji set created by Twitter and licensed under CC BY 4.0.

## Impact Statement

This paper presents work whose goal is to advance the field of Machine Learning. There are many potential societal consequences of our work, none of which we feel must be specifically highlighted here.

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

# A. Limitations

One potential limitation of FQL is that it requires numerically solving ODEs during training to minimize the distillation loss (Equation (7)). While this is not necessarily a significant speed bottleneck on both state- and pixel-based tasks in our experiments (as shown in Figure 11) since flow matching happens in the relatively low-dimensional *action* space (as opposed to image generation), we believe this may further be improved by incorporating a more advanced one-step distillation method, such as shortcut models (Frans et al., 2025). Another limitation is that it does not have a "built-in" exploration mechanism for online fine-tuning. For example, FQL does not achieve the best online fine-tuning on the `puzzle-4x4` task (Table 4), in which exploration can help avoid local optima. While we find that FQL without any additional exploration bonuses is enough to achieve strong performance on many challenging tasks (Figure 6), we believe it can be further improved by combining FQL with a more principled exploration strategy or additional specialized fine-tuning techniques, leaving them for future work. Finally, while we have demonstrated the performance of FQL on various simulated robotics tasks, we have not evaluated FQL on real-world tasks. We believe applying FQL's distillation-based policy extraction scheme to real-world robotic tasks, potentially with a pre-trained flow BC policy (Black et al., 2024), is another exciting future research direction.

# B. Implementation Details

In this section, we describe the full implementation details of FQL.

**Flow matching.** As mentioned in Section 2, we use the simplest flow-matching objective (Equation (5)) based on linear paths and uniform time sampling. We use a step count of 10 for the Euler method across all tasks, and for simplicity, we do not use sinusoidal embeddings for the time variable. See Figures 10c and 10d for ablation studies on these flow-related hyperparameters.

**Value learning.** Following standard practice in RL, we train two Q functions to improve stability. We take the mean of the two Q values for the Q loss term in the actor objective (Equation (9)). We also use the mean for the target value in the critic objective (Equation (1)) by default, but we use the minimum of the two Q values (which is often referred to as clipped double Q-learning (Fujimoto et al., 2018)) for the `adroit` and OGBench `antmaze-{large, giant}` tasks, as we find it to be slightly better. See Figure 10b for an ablation study on this choice.

**Online fine-tuning.** For offline-to-online RL, we simply add online transitions to the dataset, without distinguishing them from the offline transitions (*i.e.*, we do not use balanced sampling, unlike Lee et al. (2021b); Nakamoto et al. (2023); Ball et al. (2023)). We continue to train the components of FQL with the same objective as in offline training (Algorithm 1).

**Network architectures.** For FQL, we use $[512, 512, 512, 512]$-sized multi-layer perceptions (MLPs) for all neural networks. We apply layer normalization (Ba et al., 2016) to value networks to further stabilize training. We find that using a large enough network is especially important in navigation environments (*e.g.*, `antmaze`).

**Image processing.** For pixel-based environments, we use a smaller variant of the IMPALA encoder (Espeholt et al., 2018) and apply a random-shift augmentation with a probability of 0.5, following the official implementation of Park et al. (2025). In addition, we use frame stacking with three images, which we find to be important on some pixel-based tasks, such as `cube` and `puzzle`.

**Training and evaluation.** We train FQL with 1M gradient steps for state-based OGBench tasks and 500K steps for D4RL and pixel-based OGBench tasks, and evaluate the agent every 100K steps using 50 episodes. For OGBench, following the official evaluation scheme (Park et al., 2025), we report the average success rates across the last three evaluation epochs (800K, 900K, and 1M for state-based tasks and 300K, 400K, and 500K for pixel-based tasks). For D4RL, following Tarasov et al. (2023b), we report the performance at the last epoch. For offline-to-online RL results (Table 4), we report the performances at 1M and 2M steps.

**BC coefficient $\alpha$.** The most important hyperparameter of FQL is the BC coefficient $\alpha$ in Equation (9). We perform a hyperparameter search over $\{1000, 3000, 10000, 30000\}$ for `adroit` tasks and $\{3, 10, 30, 100, 300, 1000\}$ for the other tasks, and use the best one for each environment. We use larger values for `adroit` tasks simply because their return scale is significantly larger than that of the other tasks. We believe normalizing the Q loss as in Fujimoto & Gu (2021) would lead to more similar $\alpha$ values across different tasks. While we do not apply this normalization technique in our experiments, we recommend **enabling** Q normalization for new tasks (which is available in our official implementation) and tuning $\alpha$ starting from $\{0.03, 0.1, 0.3, 1, 3, 10\}$. See Figure 10a for an ablation study on the BC coefficient.

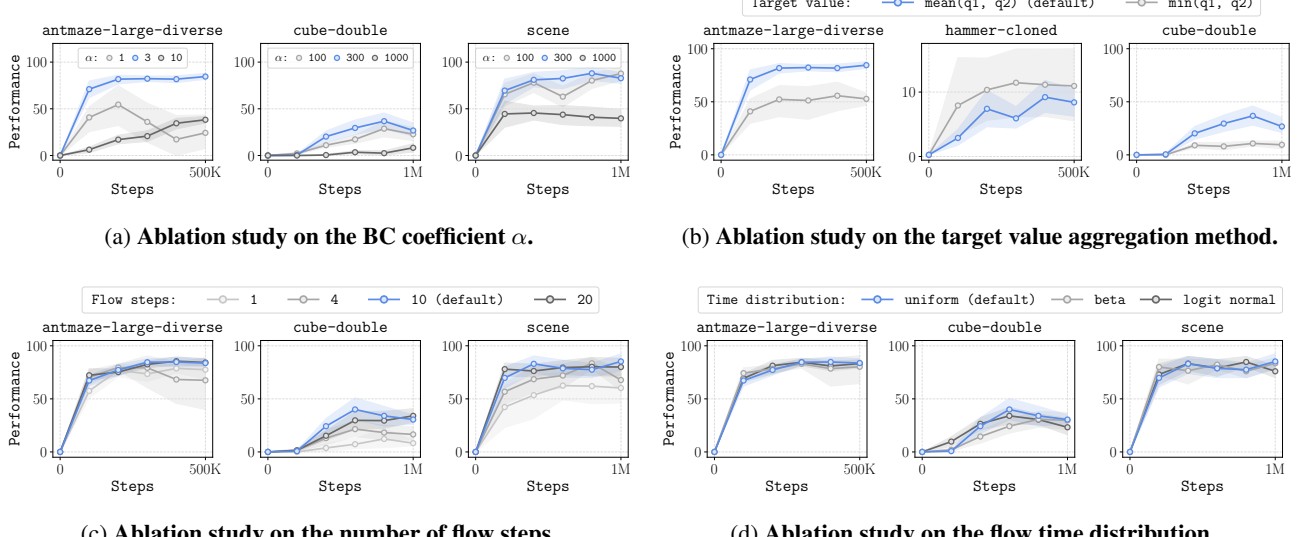

(a) **Ablation study on the BC coefficient $\alpha$.**

(b) **Ablation study on the target value aggregation method.**

(c) **Ablation study on the number of flow steps.**

(d) **Ablation study on the flow time distribution.**

*Figure 10.* **Ablation studies.** We ablate several components of FQL and study how they affect performance. The results are averaged over 8 seeds.

**Hyperparameters.** We refer to Tables 5 to 7 for the complete list of hyperparameters.

## C. Ablation Study

In this section, we ablate several components of FQL and study how they affect performance. Figure 10 shows our ablation results, where we present training curves of FQL with different hyperparameters on a representative selection of tasks.

**BC coefficient $\alpha$.** As discussed in the main paper, the BC coefficient $\alpha$ is the most important hyperparameter of FQL. Figure 10a demonstrates that $\alpha$ needs to be tuned for each task based on the suboptimality of the dataset, as is typical for most offline RL methods (Park et al., 2024a).

**Target value aggregation methods.** As discussed in Appendix B, we train two Q functions ($Q_1$ and $Q_2$) and use their mean, $(Q_1 + Q_2)/2$, for target values in the critic loss by default, but we use their minimum, $\min(Q_1, Q_2)$, for some tasks, such as adroit. We present the ablation results in Figure 10b with the BC coefficient $\alpha$ individually tuned for each ablation setting. The results show that not using clipped double Q-learning often leads to better performance, which is aligned with recent findings in online RL (Ball et al., 2023; Nauman et al., 2024; Lee et al., 2025).

**Flow steps.** To numerically solve ODEs, we use the Euler method, which requires a pre-specified number of steps. In this work, we use 10 steps for all experiments. Figure 10c shows the ablation results, which suggest that the performance is generally robust to the number of flow steps, as long as it is not too small.

**Time distributions for flow matching.** In this work, we use the uniform distribution, $\mathrm{Unif}([0, 1])$, to sample time steps for flow matching. Prior works have considered other time distributions as well. For example, Esser et al. (2024) use the logit normal distribution to emphasize intermediate steps (*i.e.*, first sample $\tilde{t}$ from the standard normal distribution, $\tilde{t} \sim \mathcal{N}(0, I)$, and then map it via the sigmoid function, $t \leftarrow 1/(1 + e^{-\tilde{t}})$), and Black et al. (2024) employ a beta distribution, $\mathrm{Beta}(1, 1.5)$, to make the flow model focus more on the initial steps. We evaluate these three strategies and report the results in Figure 10d. The results suggest that the performance is generally robust to the choice of the time distribution, and the simplest uniform distribution is often enough to achieve the best performance.

## D. Additional Results

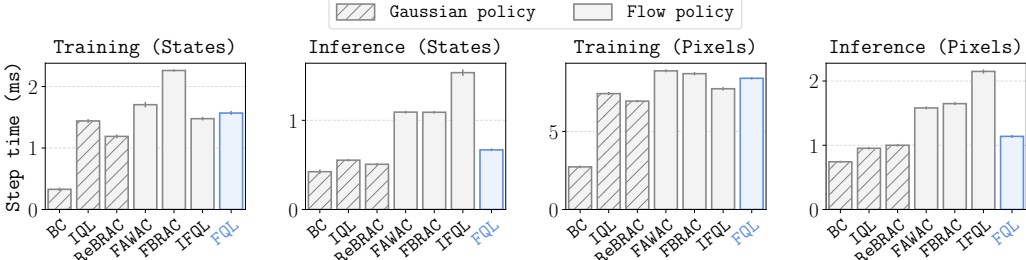

*Figure 11.* **Run time comparison.** FQL is only slightly slower than Gaussian policy-based offline RL methods, while being faster than most other flow-based methods in terms of both training and inference speeds. The run times are measured on the same machine using a single A5000 GPU, and are averaged over 8 seeds.

**Run time comparison.** Figure 11 compares the training and inference speeds of different methods on `cube-double` and `visual-cube-double`, where we consider methods implemented in the same codebase as FQL for a fair comparison. The results show that FQL achieves the best or near-best speed in terms of both training and inference among flow-based approaches. Notably, FQL is faster than FBRAC during training as it does not use potentially costly backpropagation through time, and is faster than IFQL during inference as it does not use rejection sampling.

**Full results.** We present the full per-task offline RL results in Table 3 and the full offline-to-online RL results in Table 4 and Figure 12. The results are averaged over 8 seeds (4 seeds for pixel-based tasks), and we report standard deviations after "±" in tables and 95% bootstrap confidence intervals as shaded areas in plots. In tables, we denote values at or above 95% of the best performance in bold, following OGBench (Park et al., 2025). Results without standard deviations or confidence intervals indicate that they are taken from prior work; the D4RL results of BC, IQL, ReBRAC, and Cal-QL are taken from Tarasov et al. (2023b), and the `antmaze` results of IDQL and SRPO are from Hansen-Estruch et al. (2023) and Chen et al. (2024b), respectively.

*Table 3.* **Full offline RL results.** We present the full results on the 73 OGBench and D4RL tasks. (∗) indicates the default task in each environment. The results are averaged over 8 seeds (4 seeds for pixel-based tasks) unless otherwise mentioned.

| | Gaussian Policies | | | Diffusion Policies | | | Flow Policies | | | |
|---|---|---|---|---|---|---|---|---|---|---|
| Task | BC | IQL | ReBRAC | IDQL | SRPO | CAC | FAWAC | FBRAC | IFQL | FQL |
| antmaze-large-navigate-singletask-task1-v0 (∗) | 0 ±0 | 48 ±9 | **91** ±10 | 0 ±0 | 0 ±0 | 42 ±7 | 1 ±1 | 70 ±20 | 24 ±17 | 80 ±8 |
| antmaze-large-navigate-singletask-task2-v0 | 6 ±3 | 42 ±6 | **88** ±4 | 14 ±8 | 4 ±4 | 1 ±1 | 0 ±1 | 35 ±12 | 8 ±3 | 57 ±10 |
| antmaze-large-navigate-singletask-task3-v0 | 29 ±5 | 72 ±7 | 51 ±18 | 26 ±8 | 3 ±2 | 49 ±10 | 12 ±4 | 83 ±15 | 52 ±17 | **93** ±3 |
| antmaze-large-navigate-singletask-task4-v0 | 8 ±3 | 51 ±9 | **84** ±7 | 62 ±25 | 45 ±19 | 17 ±6 | 10 ±3 | 37 ±18 | 18 ±8 | **80** ±4 |
| antmaze-large-navigate-singletask-task5-v0 | 10 ±3 | 54 ±22 | **90** ±2 | 2 ±2 | 1 ±1 | 55 ±6 | 9 ±5 | 76 ±8 | 38 ±18 | 83 ±4 |
| antmaze-giant-navigate-singletask-task1-v0 (∗) | 0 ±0 | 0 ±0 | **27** ±22 | 0 ±0 | 0 ±0 | 0 ±0 | 0 ±0 | 0 ±1 | 0 ±0 | 4 ±5 |
| antmaze-giant-navigate-singletask-task2-v0 | 0 ±0 | 1 ±1 | **16** ±17 | 0 ±0 | 0 ±0 | 0 ±0 | 0 ±0 | 4 ±7 | 0 ±0 | 9 ±7 |
| antmaze-giant-navigate-singletask-task3-v0 | 0 ±0 | 0 ±0 | **34** ±22 | 0 ±0 | 0 ±0 | 0 ±0 | 0 ±0 | 0 ±0 | 0 ±0 | 0 ±1 |
| antmaze-giant-navigate-singletask-task4-v0 | 0 ±0 | 0 ±0 | 5 ±12 | 0 ±0 | 0 ±0 | 0 ±0 | 0 ±0 | 9 ±4 | 0 ±0 | **14** ±23 |
| antmaze-giant-navigate-singletask-task5-v0 | 1 ±1 | 19 ±7 | **49** ±22 | 0 ±1 | 0 ±0 | 0 ±0 | 0 ±0 | 6 ±10 | 13 ±9 | 16 ±28 |
| humanoidmaze-medium-navigate-singletask-task1-v0 (∗) | 1 ±0 | 32 ±7 | 16 ±9 | 1 ±1 | 0 ±0 | 38 ±19 | 6 ±2 | 25 ±8 | **69** ±19 | 19 ±12 |
| humanoidmaze-medium-navigate-singletask-task2-v0 | 1 ±0 | 41 ±9 | 18 ±16 | 1 ±1 | 1 ±1 | 47 ±35 | 40 ±2 | 76 ±10 | 85 ±11 | **94** ±3 |
| humanoidmaze-medium-navigate-singletask-task3-v0 | 6 ±2 | 25 ±5 | 36 ±13 | 0 ±1 | 2 ±1 | **83** ±18 | 19 ±2 | 27 ±11 | 49 ±49 | 74 ±18 |
| humanoidmaze-medium-navigate-singletask-task4-v0 | 0 ±0 | 0 ±1 | **15** ±16 | 1 ±1 | 1 ±1 | 5 ±4 | 1 ±1 | 1 ±2 | 1 ±1 | 3 ±4 |
| humanoidmaze-medium-navigate-singletask-task5-v0 | 2 ±1 | 66 ±4 | 24 ±20 | 1 ±1 | 3 ±3 | 91 ±5 | 31 ±7 | 63 ±9 | **98** ±2 | **97** ±2 |
| humanoidmaze-large-navigate-singletask-task1-v0 (∗) | 0 ±0 | 3 ±1 | 2 ±1 | 0 ±0 | 0 ±0 | 1 ±1 | 0 ±0 | 0 ±1 | 6 ±2 | **7** ±6 |
| humanoidmaze-large-navigate-singletask-task2-v0 | **0** ±0 | **0** ±0 | **0** ±0 | **0** ±0 | **0** ±0 | **0** ±0 | **0** ±0 | **0** ±0 | **0** ±0 | **0** ±0 |
| humanoidmaze-large-navigate-singletask-task3-v0 | 1 ±1 | 7 ±3 | 8 ±4 | 3 ±1 | 1 ±1 | 2 ±3 | 1 ±1 | 10 ±2 | **48** ±10 | 11 ±7 |
| humanoidmaze-large-navigate-singletask-task4-v0 | 1 ±0 | 1 ±0 | 1 ±1 | 0 ±0 | 0 ±0 | 0 ±1 | 0 ±0 | 0 ±0 | 1 ±1 | **2** ±3 |
| humanoidmaze-large-navigate-singletask-task5-v0 | 0 ±1 | 1 ±1 | **2** ±2 | 0 ±0 | 0 ±0 | 0 ±0 | 0 ±0 | 1 ±1 | 0 ±0 | 1 ±3 |
| antsoccer-arena-navigate-singletask-task1-v0 | 2 ±1 | 14 ±5 | 0 ±0 | 44 ±12 | 2 ±1 | 1 ±3 | 22 ±2 | 17 ±3 | 61 ±25 | **77** ±4 |
| antsoccer-arena-navigate-singletask-task2-v0 | 2 ±2 | 17 ±7 | 0 ±1 | 15 ±12 | 3 ±1 | 0 ±0 | 8 ±1 | 8 ±2 | 75 ±3 | **88** ±3 |
| antsoccer-arena-navigate-singletask-task3-v0 | 0 ±0 | 6 ±4 | 0 ±0 | 0 ±0 | 0 ±0 | 8 ±19 | 11 ±5 | 16 ±3 | 14 ±22 | **61** ±6 |
| antsoccer-arena-navigate-singletask-task4-v0 (∗) | 1 ±0 | 3 ±2 | 0 ±0 | 0 ±1 | 0 ±0 | 0 ±0 | 12 ±3 | 24 ±4 | 16 ±9 | **39** ±6 |
| antsoccer-arena-navigate-singletask-task5-v0 | 0 ±0 | 2 ±2 | 0 ±0 | 0 ±0 | 0 ±0 | 0 ±0 | 9 ±2 | 15 ±4 | 0 ±1 | **36** ±9 |
| cube-single-play-singletask-task1-v0 | 10 ±5 | 88 ±3 | 89 ±5 | 95 ±2 | 89 ±7 | 77 ±28 | 81 ±9 | 73 ±33 | 79 ±4 | **97** ±2 |
| cube-single-play-singletask-task2-v0 (∗) | 3 ±1 | 85 ±8 | 92 ±4 | 96 ±3 | 82 ±16 | 80 ±30 | 81 ±9 | 83 ±13 | 73 ±3 | **97** ±2 |
| cube-single-play-singletask-task3-v0 | 9 ±3 | 91 ±5 | 93 ±3 | 99 ±1 | 96 ±2 | **98** ±1 | 87 ±4 | 82 ±12 | 88 ±4 | **98** ±2 |
| cube-single-play-singletask-task4-v0 | 2 ±1 | 73 ±6 | **92** ±3 | 93 ±4 | 70 ±18 | 91 ±2 | 79 ±6 | 79 ±20 | 79 ±6 | **94** ±3 |
| cube-single-play-singletask-task5-v0 | 3 ±3 | 78 ±9 | 87 ±8 | 90 ±6 | 61 ±12 | 80 ±20 | 78 ±10 | 76 ±33 | 77 ±7 | **93** ±3 |
| cube-double-play-singletask-task1-v0 | 8 ±3 | 27 ±5 | 45 ±6 | 39 ±19 | 7 ±6 | 21 ±8 | 21 ±7 | 47 ±11 | 35 ±9 | **61** ±9 |
| cube-double-play-singletask-task2-v0 (∗) | 0 ±0 | 1 ±1 | 7 ±3 | 16 ±10 | 0 ±0 | 2 ±2 | 2 ±1 | 22 ±12 | 9 ±5 | **36** ±6 |
| cube-double-play-singletask-task3-v0 | 0 ±0 | 0 ±0 | 4 ±1 | 17 ±8 | 0 ±1 | 3 ±1 | 1 ±1 | 4 ±2 | 8 ±5 | **22** ±5 |
| cube-double-play-singletask-task4-v0 | 0 ±0 | 0 ±0 | 1 ±1 | 0 ±1 | 0 ±0 | 0 ±1 | 0 ±0 | 0 ±1 | 1 ±1 | **5** ±2 |
| cube-double-play-singletask-task5-v0 | 0 ±0 | 4 ±3 | 4 ±2 | 1 ±1 | 0 ±0 | 3 ±2 | 2 ±1 | 2 ±2 | 17 ±6 | **19** ±10 |
| scene-play-singletask-task1-v0 | 19 ±6 | 94 ±3 | **95** ±2 | **100** ±0 | 94 ±4 | **100** ±1 | 87 ±8 | 96 ±8 | 98 ±3 | **100** ±0 |
| scene-play-singletask-task2-v0 (∗) | 1 ±1 | 12 ±3 | 50 ±13 | 33 ±14 | 2 ±2 | 50 ±40 | 18 ±8 | 46 ±10 | 0 ±0 | **76** ±9 |
| scene-play-singletask-task3-v0 | 1 ±1 | 32 ±7 | 55 ±16 | 94 ±4 | 4 ±4 | 49 ±16 | 38 ±9 | 78 ±14 | 54 ±19 | **98** ±1 |
| scene-play-singletask-task4-v0 | 2 ±2 | 0 ±1 | 3 ±3 | 4 ±3 | 0 ±0 | 0 ±0 | **6** ±1 | 4 ±4 | 0 ±0 | 5 ±1 |
| scene-play-singletask-task5-v0 | **0** ±0 | **0** ±0 | **0** ±0 | **0** ±0 | **0** ±0 | **0** ±0 | **0** ±0 | **0** ±0 | **0** ±0 | **0** ±0 |
| puzzle-3x3-play-singletask-task1-v0 | 5 ±2 | 33 ±6 | **97** ±4 | 52 ±12 | 89 ±5 | **97** ±2 | 25 ±9 | 63 ±19 | 94 ±3 | 90 ±4 |
| puzzle-3x3-play-singletask-task2-v0 | 1 ±1 | 4 ±3 | 1 ±1 | 0 ±1 | 0 ±1 | 0 ±0 | 4 ±2 | 2 ±2 | 1 ±2 | **16** ±5 |
| puzzle-3x3-play-singletask-task3-v0 | 1 ±1 | 3 ±2 | 3 ±1 | 0 ±0 | 0 ±0 | 0 ±0 | 1 ±0 | 1 ±1 | 0 ±0 | **10** ±3 |
| puzzle-3x3-play-singletask-task4-v0 (∗) | 1 ±1 | 2 ±1 | 2 ±1 | 0 ±0 | 0 ±0 | 0 ±0 | 1 ±1 | 2 ±2 | 0 ±0 | **16** ±5 |
| puzzle-3x3-play-singletask-task5-v0 | 1 ±0 | 3 ±2 | 5 ±3 | 0 ±0 | 0 ±0 | 0 ±0 | 1 ±1 | 2 ±2 | 0 ±0 | **16** ±3 |
| puzzle-4x4-play-singletask-task1-v0 | 1 ±1 | 12 ±2 | 26 ±4 | 48 ±5 | 24 ±9 | 44 ±10 | 1 ±2 | 32 ±9 | **49** ±9 | 34 ±8 |
| puzzle-4x4-play-singletask-task2-v0 | 0 ±0 | 7 ±4 | 12 ±4 | 14 ±5 | 0 ±1 | 0 ±0 | 0 ±1 | 5 ±3 | 4 ±4 | **16** ±5 |
| puzzle-4x4-play-singletask-task3-v0 | 0 ±0 | 9 ±3 | 15 ±3 | 34 ±5 | 21 ±10 | 29 ±12 | 1 ±1 | 20 ±10 | **50** ±14 | 18 ±5 |
| puzzle-4x4-play-singletask-task4-v0 (∗) | 0 ±0 | 5 ±2 | 10 ±3 | **26** ±6 | 7 ±4 | 1 ±1 | 0 ±0 | 5 ±1 | 21 ±11 | 11 ±3 |
| puzzle-4x4-play-singletask-task5-v0 | 0 ±0 | 4 ±1 | 7 ±3 | **24** ±11 | 1 ±1 | 0 ±0 | 0 ±1 | 4 ±3 | 2 ±2 | 7 ±3 |
| antmaze-umaze-v2 | 55 | 77 | **98** | 94 | 97 | 66 ±5 | 90 ±6 | **94** ±3 | 92 ±6 | **96** ±2 |
| antmaze-umaze-diverse-v2 | 47 | 54 | 84 | 80 | 82 | 66 ±11 | 55 ±7 | 82 ±9 | 62 ±12 | **89** ±5 |
| antmaze-medium-play-v2 | 0 | 66 | **90** | 84 | 81 | 49 ±24 | 52 ±12 | 77 ±7 | 56 ±15 | 78 ±7 |
| antmaze-medium-diverse-v2 | 1 | 74 | 84 | **85** | 75 | 0 ±1 | 44 ±15 | 77 ±6 | 60 ±25 | 71 ±13 |
| antmaze-large-play-v2 | 0 | 42 | 52 | 64 | 54 | 0 ±0 | 10 ±6 | 32 ±21 | 55 ±9 | **84** ±7 |
| antmaze-large-diverse-v2 | 0 | 30 | 64 | 68 | 54 | 0 ±0 | 16 ±10 | 20 ±17 | 64 ±8 | **83** ±4 |
| pen-human-v1 | 71 | 78 | **103** | 76 ±10 | 69 ±7 | 64 ±8 | 67 ±5 | 77 ±7 | 71 ±12 | 53 ±6 |
| pen-cloned-v1 | 52 | 83 | **103** | 64 ±7 | 61 ±7 | 56 ±10 | 62 ±10 | 67 ±9 | 80 ±11 | 74 ±11 |
| pen-expert-v1 | 110 | 128 | **152** | 140 ±6 | 134 ±4 | 103 ±9 | 118 ±6 | 119 ±7 | 139 ±5 | 142 ±6 |
| door-human-v1 | 2 | 3 | −0 | 6 ±2 | 3 ±3 | 5 ±2 | 2 ±1 | 4 ±2 | **7** ±2 | 0 ±0 |
| door-cloned-v1 | −0 | **3** | 0 | 0 ±0 | 0 ±0 | 1 ±0 | 0 ±1 | 0 ±0 | 2 ±2 | 2 ±1 |
| door-expert-v1 | **105** | 107 | 106 | 105 ±1 | 105 ±0 | 98 ±3 | 103 ±1 | 104 ±1 | 104 ±2 | 104 ±1 |
| hammer-human-v1 | **3** | 2 | 0 | 2 ±1 | 1 ±1 | 2 ±0 | 2 ±1 | 2 ±1 | **3** ±1 | 1 ±1 |
| hammer-cloned-v1 | 1 | 2 | 5 | 2 ±1 | 2 ±1 | 1 ±1 | 1 ±0 | 2 ±1 | 2 ±1 | **11** ±9 |
| hammer-expert-v1 | 127 | **129** | 134 | 125 ±4 | 127 ±0 | 92 ±11 | 118 ±3 | 119 ±9 | 117 ±9 | 125 ±3 |
| relocate-human-v1 | **0** | **0** | **0** | **0** ±0 | **0** ±0 | **0** ±0 | **0** ±0 | **0** ±0 | **0** ±0 | **0** ±0 |
| relocate-cloned-v1 | −0 | 0 | **2** | −0 ±0 | −0 ±0 | −0 ±0 | −0 ±0 | 1 ±1 | −0 ±0 | −0 ±0 |
| relocate-expert-v1 | 108 | 106 | 108 | 107 ±1 | 106 ±2 | 93 ±6 | 105 ±3 | **105** ±2 | 104 ±3 | 107 ±1 |
| visual-cube-single-play-singletask-task1-v0[1] | - | 70 ±12 | **83** ±6 | - | - | - | - | 55 ±8 | 49 ±7 | **81** ±12 |
| visual-cube-double-play-singletask-task1-v0[1] | - | **34** ±23 | 4 ±4 | - | - | - | - | 6 ±2 | 8 ±6 | 21 ±11 |
| visual-scene-play-singletask-task1-v0[1] | - | 97 ±2 | 98 ±4 | - | - | - | - | 46 ±4 | 86 ±10 | **98** ±3 |
| visual-puzzle-3x3-play-singletask-task1-v0[1] | - | 7 ±15 | 88 ±4 | - | - | - | - | 7 ±2 | **100** ±0 | 94 ±1 |
| visual-puzzle-4x4-play-singletask-task1-v0[1] | - | 0 ±0 | 26 ±6 | - | - | - | - | 0 ±0 | 8 ±15 | **33** ±6 |

[1] Due to the high computational cost of pixel-based tasks, we selectively benchmark 5 methods that achieve strong performance on state-based OGBench tasks.

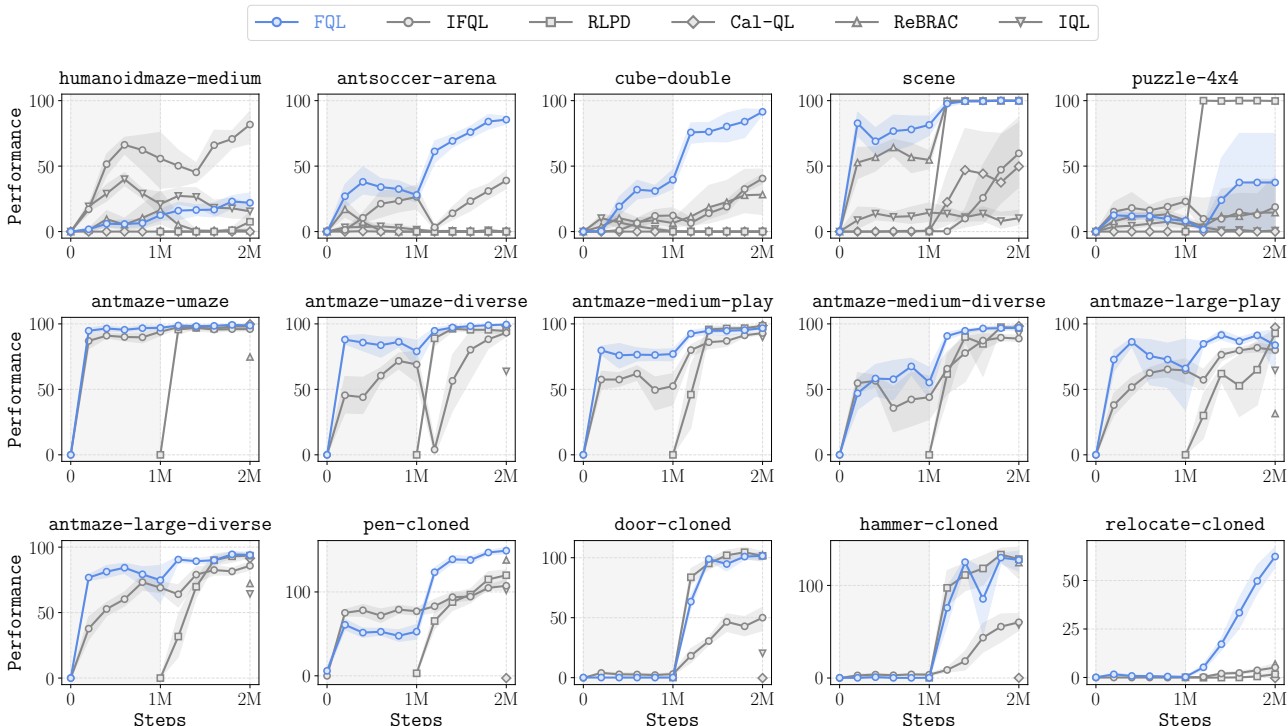

*Figure 12.* **Offline-to-online RL results.** Online fine-tuning starts at 1M steps. The results are averaged over 8 seeds unless otherwise mentioned.

*Table 4.* **Offline-to-online RL results.** The results are averaged over 8 seeds unless otherwise mentioned.

| Task | IQL | ReBRAC | Cal-QL | RLPD | IFQL | FQL |
|---|---|---|---|---|---|---|
| humanoidmaze-medium-navigate-singletask-v0 | $21_{\pm13} \to 16_{\pm8}$ | $16_{\pm20} \to 1_{\pm1}$ | $0_{\pm0} \to 0_{\pm0}$ | $0_{\pm0} \to 8_{\pm10}$ | $56_{\pm35} \to \mathbf{82}_{\pm20}$ | $12_{\pm7} \to 22_{\pm12}$ |
| antsoccer-arena-navigate-singletask-v0 | $2_{\pm1} \to 0_{\pm0}$ | $0_{\pm0} \to 0_{\pm0}$ | $0_{\pm0} \to 0_{\pm0}$ | $0_{\pm0} \to 0_{\pm0}$ | $26_{\pm15} \to 39_{\pm10}$ | $28_{\pm8} \to \mathbf{86}_{\pm5}$ |
| cube-double-play-singletask-v0 | $0_{\pm1} \to 0_{\pm0}$ | $6_{\pm5} \to 28_{\pm28}$ | $0_{\pm0} \to 0_{\pm0}$ | $0_{\pm0} \to 0_{\pm0}$ | $12_{\pm9} \to 40_{\pm10}$ | $40_{\pm11} \to \mathbf{92}_{\pm3}$ |
| scene-play-singletask-v0 | $14_{\pm11} \to 10_{\pm9}$ | $55_{\pm10} \to \mathbf{100}_{\pm0}$ | $1_{\pm2} \to 50_{\pm53}$ | $0_{\pm0} \to \mathbf{100}_{\pm0}$ | $0_{\pm1} \to 60_{\pm39}$ | $82_{\pm11} \to \mathbf{100}_{\pm1}$ |
| puzzle-4x4-play-singletask-v0 | $5_{\pm2} \to 1_{\pm1}$ | $8_{\pm4} \to 14_{\pm35}$ | $0_{\pm0} \to 0_{\pm0}$ | $0_{\pm0} \to \mathbf{100}_{\pm1}$ | $23_{\pm6} \to 19_{\pm33}$ | $8_{\pm3} \to 38_{\pm52}$ |
| antmaze-umaze-v2 | $77 \to \mathbf{96}$ | $98 \to 75$ | $77 \to \mathbf{100}$ | $0_{\pm0} \to \mathbf{98}_{\pm3}$ | $94_{\pm5} \to \mathbf{96}_{\pm2}$ | $97_{\pm2} \to 99_{\pm1}$ |
| antmaze-umaze-diverse-v2 | $60 \to 64$ | $74 \to \mathbf{98}$ | $32 \to \mathbf{98}$ | $0_{\pm0} \to 94_{\pm5}$ | $69_{\pm20} \to 93_{\pm5}$ | $79_{\pm16} \to \mathbf{100}_{\pm1}$ |
| antmaze-medium-play-v2 | $72 \to 90$ | $88 \to \mathbf{98}$ | $72 \to 99$ | $0_{\pm0} \to \mathbf{98}_{\pm2}$ | $52_{\pm19} \to 93_{\pm2}$ | $77_{\pm7} \to \mathbf{97}_{\pm2}$ |
| antmaze-medium-diverse-v2 | $64 \to 92$ | $85 \to 99$ | $62 \to \mathbf{98}$ | $0_{\pm0} \to \mathbf{97}_{\pm2}$ | $44_{\pm26} \to 89_{\pm4}$ | $55_{\pm34} \to \mathbf{97}_{\pm2}$ |
| antmaze-large-play-v2 | $38 \to 64$ | $68 \to 32$ | $32 \to \mathbf{97}$ | $0_{\pm0} \to \mathbf{93}_{\pm5}$ | $64_{\pm14} \to 80_{\pm5}$ | $66_{\pm40} \to 84_{\pm30}$ |
| antmaze-large-diverse-v2 | $27 \to 64$ | $67 \to 72$ | $44 \to 92$ | $0_{\pm0} \to \mathbf{94}_{\pm3}$ | $69_{\pm6} \to 86_{\pm5}$ | $75_{\pm24} \to \mathbf{94}_{\pm3}$ |
| pen-cloned-v1 | $84 \to 102$ | $74 \to 138$ | $-3 \to -3$ | $3_{\pm2} \to 120_{\pm10}$ | $77_{\pm7} \to 107_{\pm10}$ | $53_{\pm14} \to \mathbf{149}_{\pm6}$ |
| door-cloned-v1 | $1 \to 20$ | $0 \to \mathbf{102}$ | $-0 \to -0$ | $0_{\pm0} \to \mathbf{102}_{\pm7}$ | $3_{\pm2} \to 50_{\pm15}$ | $0_{\pm0} \to \mathbf{102}_{\pm5}$ |
| hammer-cloned-v1 | $1 \to 57$ | $7 \to \mathbf{125}$ | $0 \to 0$ | $0_{\pm0} \to \mathbf{128}_{\pm29}$ | $4_{\pm2} \to 60_{\pm14}$ | $0_{\pm0} \to \mathbf{127}_{\pm17}$ |
| relocate-cloned-v1 | $0 \to 0$ | $1 \to 7$ | $-0 \to -0$ | $0_{\pm0} \to 2_{\pm2}$ | $-0_{\pm0} \to 5_{\pm3}$ | $0_{\pm1} \to \mathbf{62}_{\pm8}$ |

# E. Experimental Details

We implement FQL and many of the baselines in JAX (Bradbury et al., 2018) on top of OGBench's reference implementations (Park et al., 2025). We provide our full implementation and exact commands to reproduce the main results of FQL at https://github.com/seohongpark/fql.

## E.1. Environments, Tasks, and Datasets

**OGBench** (Park et al., 2025). OGBench is our main benchmark, and we use 10 environments, 50 state-based tasks, and 5 pixel-based tasks from OGBench. Since OGBench was originally designed for offline goal-conditioned RL, we use the single-task variants ("-singletask") of OGBench tasks to benchmark standard reward-maximizing offline RL methods. Each OGBench environment provides five evaluation goals, each of which defines a different task (-singletask-task1 to -singletask-task5), and one of them is set to be a default task (-singletask without a suffix). Given an evaluation goal, the corresponding singletask variant labels the transitions in the dataset with a semi-sparse reward function. The

semi-sparse reward function (for the fixed task) is defined as the negative of the number of remaining subtasks at a given state. Locomotion tasks have only one subtask ("reach the goal"), and rewards are always $-1$ or $0$. Manipulation tasks usually involve more than one subtasks (*e.g.*, "open the drawer", "turn the first button's color blue", etc.), and rewards are bounded by $-n_{\text{task}}$ and $0$, where $n_{\text{task}}$ is the number of subtasks, up to $16$ in the set of environments we use. The episode ends when the agent achieves the goal.

In our experiments, we use the following $10$ state-based and $5$ pixel-based datasets (each dataset provides $5$ different tasks).

- State-based datasets
  - `antmaze-large-navigate-v0`
  - `antmaze-giant-navigate-v0`
  - `humanoidmaze-medium-navigate-v0`
  - `humanoidmaze-large-navigate-v0`
  - `antsoccer-arena-navigate-v0`
  - `cube-single-play-v0`
  - `cube-double-play-v0`
  - `scene-play-v0`
  - `puzzle-3x3-play-v0`
  - `puzzle-4x4-play-v0`
- Pixel-based datasets
  - `visual-cube-single-play-v0`
  - `visual-cube-double-play-v0`
  - `visual-scene-play-v0`
  - `visual-puzzle-3x3-play-v0`
  - `visual-puzzle-4x4-play-v0`

We choose these environments to cover diverse types of challenges. `antmaze` and `humanoidmaze` require controlling either a quadrupedal agent (with $8$ degrees of freedom) or a humanoid agent (with $21$ degrees of freedom) to reach a goal position in a given maze. `antsoccer` requires controlling a quadrupedal agent to dribble a ball to a desired location. `cube`, `scene`, and `puzzle` require manipulating diverse objects with a robot arm, where `scene` involves long-horizon control of multiple objects (up to $8$ subtasks) and `puzzle` requires combinatorial generalization. The tasks with the `visual-` prefix require pixel-based control solely from $64 \times 64 \times 3$-sized images. For dataset types, we employ the standard ones (`navigate` for locomotion and `play` for manipulation). These datasets feature high suboptimality since they consist of trajectories performing *random* tasks (*e.g.*, reaching random goals or manipulating random objects in the scene), and thus require a high degree of "stitching" capabilities. We use all of the five tasks for each state-based environment, but we use only the first task (the one labeled as `singletask-task1`) for each pixel-based environment due to high computational cost. For evaluation, we consider binary task success rates (in percentage), following the original evaluation criterion.

**D4RL** ([Fu et al., 2020](#)). To enable direct comparisons with previously reported results, we additionally employ $18$ relatively hard D4RL tasks in our experiments. We use the following $6$ `antmaze` and $12$ `adroit` tasks.

- `antmaze-umaze-v2`
- `antmaze-umaze-diverse-v2`
- `antmaze-medium-play-v2`
- `antmaze-medium-diverse-v2`
- `antmaze-large-play-v2`
- `antmaze-large-diverse-v2`
- `pen-human-v1`
- `pen-cloned-v1`
- `pen-expert-v1`
- `door-human-v1`
- `door-cloned-v1`
- `door-expert-v1`
- `hammer-human-v1`
- `hammer-cloned-v1`
- `hammer-expert-v1`

- `relocate-human-v1`
- `relocate-cloned-v1`
- `relocate-expert-v1`

D4RL `antmaze` has the same high-level objective as OGBench `antmaze`, but with different (relatively less challenging) maze layouts, datasets, and evaluation goals. `adroit` tasks (`pen`, `door`, `hammer`, and `relocate`) require dexterous manipulation with a high-dimensional (24-D) action space. We measure binary task success rates (in percentage) for `antmaze` and normalized returns for `adroit`, following the original evaluation scheme (Fu et al., 2020).

### E.2. Methods and Hyperparameters

In this work, we consider a total of 11 previous offline RL and offline-to-online RL approaches. We use the same default hyperparameters, architecture, and codebase for previous methods, unless otherwise mentioned. Also, we *individually* tune the method-specific hyperparameters of prior approaches for each environment, as described in detail below. For OGBench tasks, we tune each method on the *default* task of each environment (*i.e.*, the task corresponding to the "`-singletask`" without a task ID), and use the best hyperparameters for the other four tasks from the same environment.

**BC.** For behavioral cloning, we train a Gaussian policy with a unit standard deviation. We consider $[256, 256, 256, 256]$- and $[512, 512, 512, 512]$-sized MLPs and use the latter (which is also our default network size) for all environments.

**IQL (Kostrikov et al., 2022).** We re-implement IQL on top of the same codebase as FQL. We perform a hyperparameter search over expectile values in $\{0.7, 0.9\}$ and AWR inverse temperatures in $\{0.3, 1, 3, 10\}$. We use a fixed expectile value of $0.9$ for all environments, while the AWR inverse temperature $\alpha$ is individually tuned for each environment (Tables 6 and 7). We find that IQL tends to overfit on state-based OGBench manipulation tasks, and thus use smaller $[256, 256, 256, 256]$-sized MLPs for these state-based tasks (but not for pixel-based tasks), which we find perform better.

**ReBRAC (Tarasov et al., 2023a).** We re-implement ReBRAC on the same codebase as FQL. ReBRAC has two major hyperparameters: the actor and critic BC coefficients. We consider $\{0.003, 0.01, 0.03, 0.1, 0.3, 1\}$ for the actor BC coefficient $\alpha_1$ and $\{0, 0.001, 0.01, 0.1\}$ for the critic BC coefficient $\alpha_2$. Since actor regularization is generally (far) more important than critic regularization (Tarasov et al., 2023a), we first perform a sweep over actor BC coefficients without critic regularization, and perform a second sweep over critic BC coefficients with the best actor BC coefficient. We report the individually tuned hyperparameters in Tables 6 and 7. We use the default values for the other hyperparameters (*e.g.*, noise standard deviation, noise clipping threshold, etc.), and normalize Q values only in the actor loss, following the official implementation (Tarasov et al., 2023b).

**IDQL (Hansen-Estruch et al., 2023).** We use the official open-source implementation of IDQL. For network architectures, we use the default residual multilayer perception (MLP) (three blocks of $[256, 1024, 256]$-sized residual layers) for the behavioral diffusion policy and consider $\{[256, 256], [256, 256, 256, 256], [512, 512], [512, 512, 512, 512]\}$ for the size of the value network. We find that using 4-layer value networks in this codebase leads to unstable training, and thus choose $[512, 512]$ for OGBench locomotion tasks and $[256, 256]$ for OGBench manipulation tasks. We consider $\{0.7, 0.9\}$ for the IQL expectile value, and $\{32, 64, 128\}$ for the number of test-time action samples. We individually tune the number of action samples ($N$) for each task (Table 6), and use an IQL expectile of $0.7$ for OGBench locomotion and `adroit` tasks and $0.9$ for OGBench manipulation tasks. We use the default values for the other hyperparameters. Following the original training scheme, we train the agent for 3M steps (1.5M for value functions), three times longer than FQL's training epochs. For compatibility with our evaluation scheme, we report the average performance over 2.5M, 2.75M, and 3M steps for OGBench tasks, and the final performance for D4RL tasks.

**SRPO (Chen et al., 2024b).** For SRPO, we first used its official implementation to obtain OGBench results but were unable to achieve reasonable performance, despite initial hyperparameter sweeps. Hence, we re-implement SRPO on top of the codebase of IDQL (the closest method to SRPO), which we find to lead to better performance. We use the same tuned hyperparameters as IDQL for value learning and behavioral policy learning. For the Q coefficient ($\beta$ in Chen et al. (2024b)), we perform a hyperparameter search over $\{0.001, 0.003, 0.01, 0.03, 0.1, 0.3, 1, 3\}$ and use the best one for each environment (Table 6).

**Consistency-AC (CAC) (Ding & Jin, 2024).** We use the official open-source implementation of Consistency-AC. We consider $\{0.003, 0.01, 0.03, 0.1, 0.3, 1\}$ for the Q loss coefficient ($\eta$ in Ding & Jin (2024)) and use the best one for each environment (Table 6). For other hyperparameters for OGBench tasks, we mostly follow the default ones for D4RL `antmaze` tasks, as these are closest to OGBench tasks in that they both use sparse rewards and involve goal-reaching. Namely, we do

not normalize Q values, scale the consistency loss, and apply maximum Q backup. For D4RL `antmaze`, we re-evaluate its performances on the `-v2` tasks (the original paper uses `-v0` tasks) with the hyperparameters provided in the official implementation. For D4RL `adroit` tasks, we mainly use the default hyperparameters tuned for `adroit` but perform an additional hyperparameter sweep over Q loss coefficients in $\{0.003, 0.01, 0.03\}$ for the other tasks not used in the original paper (Table 6). For all tasks, we apply gradient clipping with a threshold of 5 and do not use online model selection to ensure a fair comparison.

**FAWR, FBRAC, and IFQL.** FAWR, FBRAC, and IFQL are implemented on top of the same codebase as FQL, sharing the same flow-matching implementation. To enable apples-to-apples comparisons, we use the same default hyperparameters as IQL for FIQL, and the same default ones as FQL for FAWR and FBRAC. However, we individually tune the policy extraction-related hyperparameters for each environment. For the inverse temperature $\alpha$ in FAWR (Equation (10)), we consider $\{0.3, 1, 3, 10\}$. For the number of test-time action samples $N$ in IFQL (Equation (11)), we consider $\{32, 64, 128\}$. For the BC coefficient $\alpha$ in FBRAC (Equation (6)), we consider $\{1000, 3000, 10000, 30000\}$ for `adroit` tasks and $\{1, 3, 10, 30, 100, 300\}$ for the other tasks. We present the task-specific hyperparameters in Tables 6 and 7.

**Cal-QL (Nakamoto et al., 2023).** We use the official implementation of Cal-QL. For the CQL regularizer coefficient $\alpha$, we consider $\{0.003, 0.01, 0.03, 0.1, 0.3, 1, 3, 10\}$ as well as its Lagrange dual variant with target action gaps $\beta$ of $\{0.2, 0.5, 0.8\}$. We use individually tuned values of these hyperparameters for different tasks (Table 7). For the network size, we consider both $[256, 256, 256, 256]$- and $[512, 512, 512, 512]$-sized MLPs, and use $[512, 512, 512, 512]$ for OGBench locomotion tasks and $[256, 256, 256, 256]$ for OGBench manipulation tasks. We also consider scaling rewards by $\{1, 3, 10\}$, and use a value of 10 to scale rewards for all tasks. We use the default values for the other hyperparameters (*e.g.*, using a mixing ratio of 0.5, taking the maximum over 10 actions when computing target values, using importance sampling for the CQL regularizer, etc.).

**RLPD (Ball et al., 2023).** We re-implement RLPD on top of the same codebase as FQL. To ensure a fair comparison with other methods, we use an update-to-data ratio of 1 and employ two Q functions. Clipped double Q-learning is only applied to D4RL `adroit` tasks, as in FQL. We do not use entropy backups, as we find it to be better.

**FQL.** See Appendix B.

We provide the complete list of hyperparameters in Table 5 and task-specific hyperparameters in Tables 6 and 7.

*Table 5.* **Hyperparameters for FQL.**

| Hyperparameter | Value |
| --- | --- |
| Learning rate | 0.0003 |
| Optimizer | Adam (Kingma & Ba, 2015) |
| Gradient steps | 1000000 (default), 500000 (D4RL, pixel-based OGBench) |
| Minibatch size | 256 |
| MLP dimensions | $[512, 512, 512, 512]$ |
| Nonlinearity | GELU (Hendrycks & Gimpel, 2016) |
| Target network smoothing coefficient | 0.005 |
| Discount factor $\gamma$ | 0.99 (default), 0.995 (`antmaze-giant`, `humanoidmaze`, `antsoccer`) |
| Image augmentation probability | 0.5 |
| Flow steps | 10 |
| Flow time sampling distribution | $\mathrm{Unif}([0, 1])$ |
| Clipped double Q-learning | False (default), True (`adroit`, `antmaze-{large, giant}-navigate`) |
| BC coefficient $\alpha$ | Tables 6 and 7 |

*Table 6.* **Task-specific hyperparameters for offline RL.** We refer to Appendix E.2 for the description for each hyperparameter variable. We individually tune these hyperparameters for each task, but in OGBench, we tune them on the default task (denoted by (∗)) and use the best hyperparameters for the other four tasks. "-" indicates that the corresponding result is taken from the prior work (or does not exist).

| Task | IQL $\alpha$ | ReBRAC $(\alpha_1, \alpha_2)$ | IDQL $N$ | SRPO $\beta$ | CAC $\eta$ | FAWAC $\alpha$ | FBRAC $\alpha$ | IFQL $N$ | FQL $\alpha$ |
|---|---|---|---|---|---|---|---|---|---|
| antmaze-large-navigate-singletask-task1-v0 (∗) | 10 | $(0.003, 0.01)$ | 32 | 0.3 | 1 | 3 | 3 | 32 | 10 |
| antmaze-large-navigate-singletask-task2-v0 | 10 | $(0.003, 0.01)$ | 32 | 0.3 | 1 | 3 | 3 | 32 | 10 |
| antmaze-large-navigate-singletask-task3-v0 | 10 | $(0.003, 0.01)$ | 32 | 0.3 | 1 | 3 | 3 | 32 | 10 |
| antmaze-large-navigate-singletask-task4-v0 | 10 | $(0.003, 0.01)$ | 32 | 0.3 | 1 | 3 | 3 | 32 | 10 |
| antmaze-large-navigate-singletask-task5-v0 | 10 | $(0.003, 0.01)$ | 32 | 0.3 | 1 | 3 | 3 | 32 | 10 |
| antmaze-giant-navigate-singletask-task1-v0 (∗) | 10 | $(0.003, 0.01)$ | 32 | 0.3 | 1 | 3 | 10 | 32 | 10 |
| antmaze-giant-navigate-singletask-task2-v0 | 10 | $(0.003, 0.01)$ | 32 | 0.3 | 1 | 3 | 10 | 32 | 10 |
| antmaze-giant-navigate-singletask-task3-v0 | 10 | $(0.003, 0.01)$ | 32 | 0.3 | 1 | 3 | 10 | 32 | 10 |
| antmaze-giant-navigate-singletask-task4-v0 | 10 | $(0.003, 0.01)$ | 32 | 0.3 | 1 | 3 | 10 | 32 | 10 |
| antmaze-giant-navigate-singletask-task5-v0 | 10 | $(0.003, 0.01)$ | 32 | 0.3 | 1 | 3 | 10 | 32 | 10 |
| humanoidmaze-medium-navigate-singletask-task1-v0 (∗) | 10 | $(0.01, 0.01)$ | 32 | 0.3 | 0.03 | 3 | 30 | 32 | 30 |
| humanoidmaze-medium-navigate-singletask-task2-v0 | 10 | $(0.01, 0.01)$ | 32 | 0.3 | 0.03 | 3 | 30 | 32 | 30 |
| humanoidmaze-medium-navigate-singletask-task3-v0 | 10 | $(0.01, 0.01)$ | 32 | 0.3 | 0.03 | 3 | 30 | 32 | 30 |
| humanoidmaze-medium-navigate-singletask-task4-v0 | 10 | $(0.01, 0.01)$ | 32 | 0.3 | 0.03 | 3 | 30 | 32 | 30 |
| humanoidmaze-medium-navigate-singletask-task5-v0 | 10 | $(0.01, 0.01)$ | 32 | 0.3 | 0.03 | 3 | 30 | 32 | 30 |
| humanoidmaze-large-navigate-singletask-task1-v0 (∗) | 10 | $(0.01, 0.01)$ | 32 | 0.3 | 1 | 3 | 30 | 32 | 30 |
| humanoidmaze-large-navigate-singletask-task2-v0 | 10 | $(0.01, 0.01)$ | 32 | 0.3 | 1 | 3 | 30 | 32 | 30 |
| humanoidmaze-large-navigate-singletask-task3-v0 | 10 | $(0.01, 0.01)$ | 32 | 0.3 | 1 | 3 | 30 | 32 | 30 |
| humanoidmaze-large-navigate-singletask-task4-v0 | 10 | $(0.01, 0.01)$ | 32 | 0.3 | 1 | 3 | 30 | 32 | 30 |
| humanoidmaze-large-navigate-singletask-task5-v0 | 10 | $(0.01, 0.01)$ | 32 | 0.3 | 1 | 3 | 30 | 32 | 30 |
| antsoccer-arena-navigate-singletask-task1-v0 | 1 | $(0.01, 0.01)$ | 32 | 0.03 | 1 | 10 | 30 | 64 | 10 |
| antsoccer-arena-navigate-singletask-task2-v0 | 1 | $(0.01, 0.01)$ | 32 | 0.03 | 1 | 10 | 30 | 64 | 10 |
| antsoccer-arena-navigate-singletask-task3-v0 | 1 | $(0.01, 0.01)$ | 32 | 0.03 | 1 | 10 | 30 | 64 | 10 |
| antsoccer-arena-navigate-singletask-task4-v0 (∗) | 1 | $(0.01, 0.01)$ | 32 | 0.03 | 1 | 10 | 30 | 64 | 10 |
| antsoccer-arena-navigate-singletask-task5-v0 | 1 | $(0.01, 0.01)$ | 32 | 0.03 | 1 | 10 | 30 | 64 | 10 |
| cube-single-play-singletask-task1-v0 | 1 | $(1, 0)$ | 32 | 0.03 | 0.003 | 1 | 100 | 32 | 300 |
| cube-single-play-singletask-task2-v0 (∗) | 1 | $(1, 0)$ | 32 | 0.03 | 0.003 | 1 | 100 | 32 | 300 |
| cube-single-play-singletask-task3-v0 | 1 | $(1, 0)$ | 32 | 0.03 | 0.003 | 1 | 100 | 32 | 300 |
| cube-single-play-singletask-task4-v0 | 1 | $(1, 0)$ | 32 | 0.03 | 0.003 | 1 | 100 | 32 | 300 |
| cube-single-play-singletask-task5-v0 | 1 | $(1, 0)$ | 32 | 0.03 | 0.003 | 1 | 100 | 32 | 300 |
| cube-double-play-singletask-task1-v0 | 0.3 | $(0.1, 0)$ | 32 | 0.1 | 0.3 | 0.3 | 100 | 32 | 300 |
| cube-double-play-singletask-task2-v0 (∗) | 0.3 | $(0.1, 0)$ | 32 | 0.1 | 0.3 | 0.3 | 100 | 32 | 300 |
| cube-double-play-singletask-task3-v0 | 0.3 | $(0.1, 0)$ | 32 | 0.1 | 0.3 | 0.3 | 100 | 32 | 300 |
| cube-double-play-singletask-task4-v0 | 0.3 | $(0.1, 0)$ | 32 | 0.1 | 0.3 | 0.3 | 100 | 32 | 300 |
| cube-double-play-singletask-task5-v0 | 0.3 | $(0.1, 0)$ | 32 | 0.1 | 0.3 | 0.3 | 100 | 32 | 300 |
| scene-play-singletask-task1-v0 | 10 | $(0.1, 0.01)$ | 32 | 0.1 | 0.3 | 0.3 | 100 | 32 | 300 |
| scene-play-singletask-task2-v0 (∗) | 10 | $(0.1, 0.01)$ | 32 | 0.1 | 0.3 | 0.3 | 100 | 32 | 300 |
| scene-play-singletask-task3-v0 | 10 | $(0.1, 0.01)$ | 32 | 0.1 | 0.3 | 0.3 | 100 | 32 | 300 |
| scene-play-singletask-task4-v0 | 10 | $(0.1, 0.01)$ | 32 | 0.1 | 0.3 | 0.3 | 100 | 32 | 300 |
| scene-play-singletask-task5-v0 | 10 | $(0.1, 0.01)$ | 32 | 0.1 | 0.3 | 0.3 | 100 | 32 | 300 |
| puzzle-3x3-play-singletask-task1-v0 | 10 | $(0.3, 0.01)$ | 32 | 0.1 | 0.01 | 0.3 | 100 | 32 | 1000 |
| puzzle-3x3-play-singletask-task2-v0 | 10 | $(0.3, 0.01)$ | 32 | 0.1 | 0.01 | 0.3 | 100 | 32 | 1000 |
| puzzle-3x3-play-singletask-task3-v0 | 10 | $(0.3, 0.01)$ | 32 | 0.1 | 0.01 | 0.3 | 100 | 32 | 1000 |
| puzzle-3x3-play-singletask-task4-v0 (∗) | 10 | $(0.3, 0.01)$ | 32 | 0.1 | 0.01 | 0.3 | 100 | 32 | 1000 |
| puzzle-3x3-play-singletask-task5-v0 | 10 | $(0.3, 0.01)$ | 32 | 0.1 | 0.01 | 0.3 | 100 | 32 | 1000 |
| puzzle-4x4-play-singletask-task1-v0 | 3 | $(0.3, 0.01)$ | 32 | 0.1 | 0.01 | 0.3 | 300 | 32 | 1000 |
| puzzle-4x4-play-singletask-task2-v0 | 3 | $(0.3, 0.01)$ | 32 | 0.1 | 0.01 | 0.3 | 300 | 32 | 1000 |
| puzzle-4x4-play-singletask-task3-v0 | 3 | $(0.3, 0.01)$ | 32 | 0.1 | 0.01 | 0.3 | 300 | 32 | 1000 |
| puzzle-4x4-play-singletask-task4-v0 (∗) | 3 | $(0.3, 0.01)$ | 32 | 0.1 | 0.01 | 0.3 | 300 | 32 | 1000 |
| puzzle-4x4-play-singletask-task5-v0 | 3 | $(0.3, 0.01)$ | 32 | 0.1 | 0.01 | 0.3 | 300 | 32 | 1000 |
| antmaze-umaze-v2 | - | - | - | - | 0.01 | 3 | 10 | 32 | 10 |
| antmaze-umaze-diverse-v2 | - | - | - | - | 0.01 | 3 | 10 | 32 | 10 |
| antmaze-medium-play-v2 | - | - | - | - | 0.01 | 3 | 10 | 32 | 10 |
| antmaze-medium-diverse-v2 | - | - | - | - | 0.01 | 3 | 10 | 32 | 10 |
| antmaze-large-play-v2 | - | - | - | - | 4.5 | 3 | 1 | 32 | 3 |
| antmaze-large-diverse-v2 | - | - | - | - | 3.5 | 3 | 1 | 32 | 3 |
| pen-human-v1 | - | - | 32 | 0.03 | 0.003 | 0.03 | 30000 | 32 | 10000 |
| pen-cloned-v1 | - | - | 32 | 0.1 | 0.003 | 0.3 | 10000 | 32 | 10000 |
| pen-expert-v1 | - | - | 32 | 0.1 | 0.03 | 0.1 | 30000 | 32 | 3000 |
| door-human-v1 | - | - | 32 | 0.01 | 0.03 | 1 | 30000 | 32 | 30000 |
| door-cloned-v1 | - | - | 32 | 0.03 | 0.03 | 1 | 10000 | 128 | 30000 |
| door-expert-v1 | - | - | 32 | 0.01 | 0.03 | 3 | 30000 | 32 | 30000 |
| hammer-human-v1 | - | - | 128 | 0.1 | 0.03 | 3 | 30000 | 32 | 30000 |
| hammer-cloned-v1 | - | - | 32 | 0.1 | 0.003 | 0.03 | 10000 | 32 | 10000 |
| hammer-expert-v1 | - | - | 32 | 0.03 | 0.03 | 3 | 30000 | 32 | 30000 |
| relocate-human-v1 | - | - | 32 | 0.03 | 0.01 | 0.3 | 30000 | 128 | 10000 |
| relocate-cloned-v1 | - | - | 64 | 0.03 | 0.01 | 0.1 | 3000 | 32 | 30000 |
| relocate-expert-v1 | - | - | 32 | 0.01 | 0.003 | 1 | 30000 | 32 | 30000 |
| visual-cube-single-play-singletask-task1-v0 | 1 | $(1, 0)$ | - | - | - | - | 100 | 32 | 300 |
| visual-cube-double-play-singletask-task1-v0 | 0.3 | $(0.1, 0)$ | - | - | - | - | 100 | 32 | 100 |
| visual-scene-play-singletask-task1-v0 | 10 | $(0.1, 0.01)$ | - | - | - | - | 100 | 32 | 100 |
| visual-puzzle-3x3-play-singletask-task1-v0 | 10 | $(0.3, 0.01)$ | - | - | - | - | 100 | 32 | 300 |
| visual-puzzle-4x4-play-singletask-task1-v0 | 3 | $(0.3, 0.01)$ | - | - | - | - | 300 | 32 | 300 |

*Table 7.* **Task-specific hyperparameters for offline-to-online RL.** We refer to Appendix E.2 for the description for each hyperparameter variable. We individually tune these hyperparameters for each task, and "-" indicates that the corresponding result is taken from the prior work.

| Task | IQL $\alpha$ | ReBRAC $(\alpha_1, \alpha_2)$ | Cal-QL $(\alpha, \beta)$ | IFQL $N$ | FQL $\alpha$ |
|---|---|---|---|---|---|
| humanoidmaze-medium-navigate-singletask-v0 | 10 | $(0.01, 0.01)$ | $(-, 0.8)$ | 32 | 100 |
| antsoccer-arena-navigate-singletask-v0 | 1 | $(0.01, 0.01)$ | $(-, 0.2)$ | 64 | 30 |
| cube-double-play-singletask-v0 | 0.3 | $(0.1, 0)$ | $(0.01, -)$ | 32 | 300 |
| scene-play-singletask-v0 | 10 | $(0.1, 0.01)$ | $(0.01, -)$ | 32 | 300 |
| puzzle-4x4-play-singletask-v0 | 3 | $(0.3, 0.01)$ | $(0.003, -)$ | 32 | 1000 |
| antmaze-umaze-v2 | - | - | - | 32 | 10 |
| antmaze-umaze-diverse-v2 | - | - | - | 32 | 10 |
| antmaze-medium-play-v2 | - | - | - | 32 | 10 |
| antmaze-medium-diverse-v2 | - | - | - | 32 | 10 |
| antmaze-large-play-v2 | - | - | - | 32 | 3 |
| antmaze-large-diverse-v2 | - | - | - | 32 | 3 |
| pen-cloned-v1 | - | - | - | 128 | 1000 |
| door-cloned-v1 | - | - | - | 128 | 1000 |
| hammer-cloned-v1 | - | - | - | 128 | 1000 |
| relocate-cloned-v1 | - | - | - | 128 | 10000 |

