# OpenReview forum: "Flow Q-Learning"
_ICML.cc/2025/Conference — ICML 2025 poster_

### Official Review · Reviewer_NB9x · 2025-02-26

**Overall Recommendation:** 3

**Summary:**

This paper proposes using flow models to tackle offline RL tasks. To leverage the Q-function for guiding the learning of the flow-based policy while avoiding the computational cost of multiple backpropagations, this paper proposes learning a one-step action generation policy through a flow policy constrained distillation loss.

**Claims And Evidence:**

Most of the claims in this paper are well supported, but some parts remain difficult to understand. For details, please refer to the questions and comments.

**Essential References Not Discussed:**

None

**Experimental Designs Or Analyses:**

Please refer to the questions and comments for experimental concerns.

**Methods And Evaluation Criteria:**

This paper compares several classic offline RL methods.

**Other Comments Or Suggestions:**

1. In line 110: To make the description clearer and more intuitive, I suggest the authors directly use notations like $a^1$ and a^0 to represent actions at different time points.
2. In line 139 right column: Whether \mu_w serves as deterministic policy? In other words, you want to distill a deterministic policy that maximizes the Q function and, at the same time, minimizes the output discrepancy with flow policy $\mu_\theta$.
3. I suggest to add the parameters comparison between $\mu_w$, $\mu_\theta$, and Gaussian policies.
4. Compared to previous baselines, the key difference in the method used in this paper lies in the policy constraint. Specifically, it measures the distance between a deterministic policy and an expressive policy, whereas previous Gaussian policy methods impose constraints on the state-to-action mapping learned from the dataset, and the diffusion policies measure the distance between two expressive policies. It is better to understand this paper by discussing the differences among these types of applying policy constraints.

**Other Strengths And Weaknesses:**

Please refer to the questions and comments for review concerns.

**Questions For Authors:**

1. Is the analysis in the Remark redundant? It does not seem to contribute much to the understanding of the proposed method.
2. Would using a better ODE solver lead to improved performance? Would using a different marginal probability path result in better performance?

**Relation To Broader Scientific Literature:**

In recent years, diffusion-based RL methods have shown great potential in offline RL tasks. As a more general generative model compared to diffusion models, applying flow models to RL is a promising research direction.

**Theoretical Claims:**

I have read the theoretical parts related to the paper.

---

> ### Author Rebuttal · Authors · 2025-03-27
>
> Thank you for the detailed review and constructive feedback on this work. We especially appreciate the clarification questions about deterministic policies and policy constraints, as well as several helpful suggestions. We also conducted an additional ablation study on ODE solvers. Please find our response below.
>
> ---
>
> * **Would a better ODE solver/different marginal probability path improve performance?**
>
> Thank you for the question. Following the suggestion, we conducted an additional ablation study to compare different ODE solvers for FQL. We consider three ODE solvers in this experiment: (1) the Euler method (**default**), (2) the midpoint method, and (3) the Runge-Kutta (RK4) method. The table below compares the final performance of these ODE solvers on $3$ OGBench tasks ($8$ seeds, $\pm$ denotes standard deviations):
>
> | Task | Euler (Default) | Midpoint | RK4 |
> |-|-|-|-|
> | $\texttt{antmaze-large}$ | $76 \pm 23$ | $83 \pm 6$ | $82 \pm 4$ |
> | $\texttt{cube-double}$ | $56 \pm 16$ | $54 \pm 17$ | $7 \pm 20$ |
> | $\texttt{scene}$ | $87 \pm 8$ | $83 \pm 12$ | $83 \pm 5$ |
>
> The results suggest that the performance of FQL is generally robust to the choice of ODE solver, except that RK4 leads to somewhat unstable performance on $\texttt{cube-double}$. While we have not tested different marginal probability paths, we expect similar results, given that the performance of FQL is generally robust to other flow-related hyperparameters (Figure 9). That said, we believe a better ODE solver or marginal path sampler might further improve performance in more complex tasks and datasets.
>
> * **About different kinds of policy constraints.**
>
> As the reviewer correctly pointed out, Gaussian behavior-constrained actor-critic methods (e.g., ReBRAC) minimize the distance between the RL policy's actions and dataset actions, while FQL minimizes the distance between the RL policy's actions and the BC policy's actions. They both serve the same role of a behavioral regularizer. The main difference is that ReBRAC *directly* imposes a *Gaussian*-based constraint, while FQL *indirectly* imposes a *flow*-based constraint. We note that FBRAC in our experiment lies precisely between the two: it *directly* imposes a *flow*-based behavioral constraint.
>
> We have empirically compared these $3$ policy extraction schemes (ReBRAC, FBRAC, and FQL) in Table 2 of the paper. The result shows that our indirect flow-based behavioral constraint (FQL) indeed leads to better performance than both the direct Gaussian-based behavioral constraint (ReBRAC) and direct flow-based behavioral constraint (FBRAC). Following the suggestion, we will further clarify the differences between these policy constraints in the final version of the paper.
>
> * **Is $\mu_\omega$ in L139 a deterministic policy?**
>
> As mentioned in the "Notational Warning" paragraph (L126), $\mu_\omega(s, z)$ is a deterministic function. However, this does **not** mean that we perform distillation into a deterministic policy: even though $\mu_\omega(s, z)$ is a deterministic function, since $z$ is a random variable (sampled from $\mathcal{N}(0, I)$), it serves as a *stochastic* policy $\pi_\omega(a \mid s)$ when marginalizing out $z$. As a result, the one-step policy clones the flow BC policy **for each $\mathbf{z}$** through *deterministic distillation*, while maximizing the Q function with *stochastic* actions (Eq. (8)). While we (partly) explained this subtlety in the "Notational Warning" paragraph, we will further clarify this point in the final draft to prevent any potential confusion.
>
> * **About the remark box.**
>
> The purpose of the remark box is to provide further theoretical insight into the policy constraint of FQL and to discuss its relation to previous offline RL methods (TD3+BC, AWAC, CQL, etc.). We believe this section can be safely skipped if the reader is mainly interested in empirical results and methodology (which is the main reason we formatted this discussion with a separate box).
>
> * **Parameter comparisons between policies.**
>
> To clarify, we used the same [512, 512, 512, 512]-sized MLPs *all* networks (including flow policies, one-step policies, and Gaussian policies; Table 5), unless otherwise noted, and they have almost identical numbers of parameters. There are some exceptions (e.g., IDQL), but we used smaller networks *only* when the smaller ones performed better than the default-sized ones. We fully described the way we chose these hyperparameters in Appendix F.2. We will further clarify this point in the final draft.
>
> * **Notational suggestions.**
>
> Thanks for the helpful suggestions! We will incorporate them in the camera-ready version.
>
> ---
>
> We would like to thank the reviewer again for raising important questions about FQL. We believe the additional results and clarifications have significantly improved the quality of the paper. Please let us know if you have any additional questions or concerns. If we have addressed your concerns, would you consider raising your rating?

---

> > ### Comment · Reviewer_NB9x · 2025-04-04
> >
> > I thank the authors for their clarification and additional experiments. Most of my concerns are addressed. I tend to keep the current review evaluation.

---

### Official Review · Reviewer_HKD1 · 2025-03-14

**Overall Recommendation:** 3

**Summary:**

The paper introduces Flow Q-learning (FQL), an offline RL method that combines flow-matching policies with Q-learning to address challenges in modeling complex action distributions. FQL uses two components: (1) an expressive flow-matching policy trained via behavioral cloning (BC) to capture multimodal dataset actions, and (2) a separate one-step policy trained with Q-learning to maximize values while distilling knowledge from the flow model. This decoupling avoids unstable recursive backpropagation through iterative flow steps and eliminates costly iterative action generation during evaluation.

**Claims And Evidence:**

Yes

**Essential References Not Discussed:**

The method proposed by the authors bears a resemblance to reward distillation in alignment for image diffusion, where a few-step model is distilled while simultaneously maximizing the reward.

[1] Reward Guided Latent Consistency Distillation​

[2] DOLLAR: Few-Step Video Generation via Distillation and Latent Reward Optimization

**Experimental Designs Or Analyses:**

D4RL

**Methods And Evaluation Criteria:**

Yes

**Other Comments Or Suggestions:**

No

**Other Strengths And Weaknesses:**

No

**Questions For Authors:**

Could you provide more details about the inference step for the BC Flow Policy? Specifically, I would like to know whether the BC Flow Policy incorporates classifier-free guidance (CFG) during its inference process. Additionally, I’m curious if the one-step policy also utilizes CFG.

**Relation To Broader Scientific Literature:**

The article provides a detailed discussion on the relevant literature concerning RL with diffusion and flow models.

**Theoretical Claims:**

No

---

> ### Author Rebuttal · Authors · 2025-03-27
>
> Thank you for the detailed review and constructive feedback on this work. We especially appreciate your pointing out related work in different domains. Please find our response below.
>
> ---
>
> * **Inference details of policies.**
>
> Thanks for asking this clarification question! In FQL, *neither* the BC flow policy nor the one-step policy uses CFG (or CG), as there are no separate class labels in our problem setting. More specifically, the one-step policy $\mu_\omega(s, z)$ is simply a standard feedforward neural network (so no iterative sampling procedure is needed), and the BC flow policy $\mu_\theta(s, z)$ is based on the standard Euler method for ODE solving. At evaluation (inference) time, only the one-step policy is used (L160), and no iterative process is involved. We refer to the algorithm box in the main text (Algorithm 1) as well as L679-L688 in Appendix C for the full description of the sampling procedure.
>
> * **Related works on image diffusion models.**
>
> Thanks for pointing out these relevant works in image diffusion models! We will cite and discuss them in the camera-ready version.
>
> ---
>
> We would like to thank you again for raising clarification questions and suggesting several related works. Please feel free to let us know if you have any additional questions or concerns. If we have addressed your concerns, would you consider raising your rating?

---

### Official Review · Reviewer_4ovk · 2025-03-17

**Overall Recommendation:** 3

**Summary:**

This paper propose Flow Q-learning, and offline reinforcement learning that integrates expressive flow-matching policies for modeling complex action distributions.

## Update after rebuttal:
I have read the rebuttal and the discussions from other reviewers. I am maintaining my score.

**Claims And Evidence:**

The paper identifies a clear challenge in offline RL related to using flow or diffusion policies. It presents an elegant solution: training an expressive one-step policy separately from the iterative flow policy, which is both theoretically sound and empirically effective.

**Essential References Not Discussed:**

Not applicable.

**Experimental Designs Or Analyses:**

Extensive experiments are performed, demonstrating consistently strong performance across benchmark tasks.

**Methods And Evaluation Criteria:**

The approach is grounded in existing behavior-regularized actor-critic frameworks but innovatively avoids costly BPTT during RL training.

**Other Comments Or Suggestions:**

N/A

**Other Strengths And Weaknesses:**

N/A

**Questions For Authors:**

N/A

**Relation To Broader Scientific Literature:**

The key contributions of this paper are related to offline reinforcement learning, generative modeling (specifically flow matching), and policy extraction techniques.

**Theoretical Claims:**

The derivation relating the distillation loss to a Wasserstein behavioral regularizer provides additional insight and theoretical justification for why the proposed method might perform better.

---

> ### Author Rebuttal · Authors · 2025-03-27
>
> Thank you for the positive feedback about this work! We would be happy to address any additional questions or concerns you may have, so please feel free to let us know. If there are no further concerns or questions, would you consider raising your rating?

---

### Official Review · Reviewer_YTug · 2025-03-23

**Overall Recommendation:** 4

**Summary:**

This paper proposes the offline-RL method Flow-Q-Learning (FQL) which leverages  an expressive flow-based generative model for modeling the action distribution while avoiding common issues such as unstable backprop through each time-step or less efficient re-weighting schemes. This is achieved by introducing a so-called “1-step” policy which is used to minimize the Q function while being regularized to match the behavior behavior cloning flow-model that learns the offline transitions from data. This 1-step model can then be evaluated efficiently and shows increased performance across many OGBench and D4RL tasks compared to previous offline methods.

**Claims And Evidence:**

-The paper claims that FQL outperforms previous offline RL methods on a wide range of tasks and in particular this type of behavior constrained policy extraction method is better than previous flow/diffusion based extraction schemes like advantage weighted flow-matching.

-They claim that this 1-step policy preserves the expressivity of the underlying action distribution learned by the BC flow-matching model.

-Training FQL is more computationally efficient and effective than related flow-based policy extraction approaches.

I think that all claims are pretty well substantiated by the results experimentally except for the 1-step policy expressiveness other than the remark about how the distillation loss is an upper bound for the W2 distance between the policies. For this reason it makes sense that the parameter \alpha needs to be tuned carefully.

**Essential References Not Discussed:**

I don't feel too strongly about this, but It might be useful to discuss how offline RL and FQL deal with more realistic data sources /real world data with noise. Perhaps this one is suitable:

Zhou 2022, Real World Offline Reinforcement Learning with Realistic Data Source

**Experimental Designs Or Analyses:**

The experimental section to be very thorough when comparing against previous offline RL approaches. I found the QA format of the discussion engaging and informative to read from a practitioner's standpoint.

It would be nice to better understand the limitations of FQL either in scalability due to the simulation-based loss (atleast it requires a forward pass of the ODE) or task expressiveness and when it does not compare well to simple Gaussian policies.

**Methods And Evaluation Criteria:**

The proposed method is very practical with the application of offline RL in mind. All design choices are well supported by the OGBench and D4RL benchmarks.

**Other Comments Or Suggestions:**

This is minor, but I would find the loss function and the algorithm more readable if instead of the notation “a^\pi” you used “a^\omega” or “a^\theta” to help distinguish which policy the action is sampled from.

In the Remark, It seems like \pi^\theta and \nu^\theta are the same thing?

**Other Strengths And Weaknesses:**

Strengths:
-The methodology is clear and simple, justified from the successes and failures of previous works.

-Strong empirical results and ablations

-Very clear and informative discussion of previous approaches which I believe is useful to practitioners.

Weaknesses:
-Some aspects of FQL aren’t fully addressed such as scalability to state-action dimension and real world data sources.

-Its not very clear how expressive the 1-step policy and how it trades off BC policy distribution and the bias/titling of the distribution towards the Q function. There is probably a trade-off between staying close to the expressive flow-model state-action representation and decreasing the Q-function more quickly.

**Questions For Authors:**

1. Is there a way to clearly demonstrate the expressiveness of the 1-step policy and how it trades off BC policy distribution and the bias/titling of the distribution towards the Q function? There is probably a trade-off between staying close to the expressive flow-model state-action representation and decreasing the Q-function more quickly.

2. How does FQL scale with state-action dimension and real world data sources which are potentially noisy?

**Relation To Broader Scientific Literature:**

The paper contextualizes its contribution quite well with existing research integrating generative models (diffusion, flow) with RL. It spends quite a lot of time distinguishing the mechanisms used from works like Diffusion-QL, IDQL, and CAC and why FQL might be more performant.

**Theoretical Claims:**

This paper does not really make any theoretical claims except for the connection between the W2 distance and the distillation regularizer. I didn't check this claim carefully, but the authors do ablate the parameter alpha.

---

> ### Author Rebuttal · Authors · 2025-03-27
>
> Thank you for the highly detailed review and constructive feedback about this work. We especially appreciate your question about the expressivity of the one-step policy, for which we conducted an additional experiment. Please find our response below.
>
> ---
>
> * **Is the one-step policy expressive enough?**
>
> Thanks for asking this valid question. To empirically address it, we performed a controlled ablation study to compare the expressivity of (1) Gaussian BC, (2) full flow BC, and (3) one-step distilled BC policies. Here, we evaluate their *BC* performance (not RL performance) to solely focus on their expressivity without being confounded by different policy extraction strategies (in particular, note that there's no clear, straightforward way to train a full flow policy with RL).
>
> Specifically, we trained these three types of policies on $4$ goal-conditioned OGBench tasks, and measured their goal-conditioned BC performance. To ensure a fair comparison, we used the same architecture and common hyperparameters, and ablated only the policy class. The table below shows the mean and standard deviation across 8 seeds at 1M gradient steps (200K steps for $\texttt{cube-single}$ due to overfitting).
>
> | Task | Gaussian | Full Flow | One-Step Distillation |
> |-|-|-|-|
> | $\texttt{antmaze-medium}$ | $77 \pm 3$ | $82 \pm 6$ | $85 \pm 6$ |
> | $\texttt{cube-single}$ | $54 \pm 17$ | $81 \pm 15$ | $90 \pm 3$ |
> | $\texttt{cube-double}$ | $4 \pm 3$ | $15 \pm 9$ | $19 \pm 9$ |
> | $\texttt{scene}$ | $13 \pm 3$ | $34 \pm 5$ | $34 \pm 8$ |
>
> The results show that one-step distilled policies achieve nearly identical performance to full flow policies on these tasks, and both flow variants generally outperform Gaussian policies.
>
> Overall, we expect that one-step policies should be expressive enough to model complex action distributions in many practical scenarios, especially considering that one-step flow models can generate highly realistic samples [1, 2] even for high-dimensional image generation.
>
> [1] Liu et al., Flow Straight and Fast: Learning to Generate and Transfer Data with Rectified Flow (2023). \
> [2] Frans et al., One Step Diffusion via Shortcut Models (2025).
>
> * **Potential trade-off by using a one-step policy.**
>
> As shown in the table above, one-step policies often have a very similar expressivity to full flow policies (at least on our benchmark tasks). Hence, we expect the "trade-off cost" of using a one-step policy to be generally marginal. That said, this trade-off might come into play if the task requires *highly* precise control and thus the full expressivity of iterative generative models. In this case, one solution would be to relax FQL to train a *few*-step distillation policy to strike a balance between precision and the number of recursive backpropagations. We leave this extension of FQL for future work.
>
> * **Scalability to higher state-action dimensions and real-world data sources.**
>
> Thanks for the question. We would like to first note that we have evaluated FQL on a diverse set of high-dimensional benchmark tasks (e.g., $\texttt{visual-*}$ tasks require *pixel*-based control, $\texttt{humanoidmaze}$ requires $21$-DoF whole-body control, and $\texttt{adroit}$ tasks require $24$-DoF dexterous manipulation). Moreover, the OGBench tasks used in this work are generally more complex than the D4RL tasks used in prior work, and they feature noisy, non-Markovian, multi-modal action distributions that (partly) resemble real-world datasets. That said, we did not evaluate FQL on real robotics data, as this work focused more on the algorithmic side of the method. We believe applying FQL to real robots with pre-trained VLA BC flow models [3] is a particularly exciting direction for future research. We will mention the lack of real-world experiments as a limitation (and discuss relevant work) in the final version of the paper.
>
> [3] Black et al., π0: A Vision-Language-Action Flow Model for General Robot Control (2024).
>
> * **Notational suggestions (e.g., $a^\theta$ instead of $a^\pi$) and minor clarifications.**
>
> Thanks for the helpful suggestions! We will incorporate them into the camera-ready version. In the remark box, as the reviewer correctly pointed out, $\pi^\theta$ and $\nu^\theta$ correspond to the same distribution. Our original intention was to distinguish $\nu^\theta$ as a probability measure, but in hindsight, we feel that this distinction is unnecessary, and we will revise the paper to use the same $\pi^\theta$ notation in the remark box.
>
> ---
>
> We would like to thank you again for raising important questions about FQL, and we believe the additional results and clarifications have strengthened the paper. Please let us know if you have any additional concerns or questions.

---

> > ### Comment · Reviewer_YTug · 2025-04-07
> >
> > I want to thank the authors for their follow up and feedback. My concerns have been addressed and I maintain my positive rating.

---

### Decision · Program_Chairs · 2025-05-01

**Decision:**

Accept (poster)

**Comment:**

The paper presents flow Q-learning (FQL), a simple and effective offline RL method using an expressive flow-matching policy to model complex action distributions in data, addressing training challenges by training a one-step policy instead of an iterative flow policy. After the rebuttal discussion, the reviewers consistently think this paper can be accepted. I personally like this paper as it tackles the fundamental challenge for modeling complex action distribution through a simple flow-based algorithm.